# Spaceborne infrared imagery for early detection of Weddell Polynya opening

Céline Heuzé[1], Lu Zhou[1], Martin Mohrmann[2], and Adriano Lemos[1]

[1]Department of Earth Sciences, University of Gothenburg, Gothenburg, Sweden
[2]Department of Marine Sciences, University of Gothenburg, Gothenburg, Sweden

**Correspondence:** Céline Heuzé (celine.heuze@gu.se)

**Abstract.** Knowing when sea ice will open is crucial, notably for scientific deployments. This was particularly obvious when the Weddell Polynya, a large opening in the winter Southern Ocean sea ice, unexpectedly re-appeared in 2016. As no precursor had been detected, observations were limited to chance autonomous sensors, and the exact cause of the opening could not be determined accurately. We investigate here whether the signature of the vertical ocean motions or that of the leads, which ultimately re-open the polynya, are detectable in spaceborne infrared temperature before the polynya opens. From the full historical sea ice concentration record, we find 30 polynyas since 1980. Then, using the full time series of the spaceborne infrared Advanced Very High Resolution Radiometer, we determine that these events can be detected in the two weeks before the polynya opens as a reduction in the variance of the data. For the three commonly used infrared brightness temperature bands, the 15-day sum and 15-day standard deviation of their area-median and -maximum are systematically lower than the climatology when a polynya will open. Moreover, by comparing the infrared brightness temperature to atmospheric reanalysis, hydrographic mooring data, and autonomous profilers, we find that temporal oscillations in one band and decrease in the difference between bands may be used as proxies for upwelling of warm water and presence of leads respectively, albeit with caution. Therefore, although infrared data are strongly limited by their horizontal resolution and sensitivity to clouds, they could be used for studying ocean or atmosphere preconditioning of polynyas in the historical record.

## 1 Introduction

Global changes in the sea ice cover, of which the continuous decrease in summer Arctic sea ice since satellite observations began in the 1970s is the most dramatic example (Stocker et al., 2014; Notz and Stroeve, 2016), lead to a development in exploitation of ice-infested waters in both hemispheres (Meier et al., 2014; Schillat et al., 2016). Consequently, for planning purposes, early detection of sea ice opening is urgently needed. Polar researchers would also benefit from such product. In 2016, the most famous opening in the Antarctic sea ice, the Weddell Polynya (Carsey, 1980), re-appeared for the first time in over a decade (Swart et al., 2018). Luckily, two autonomous profilers were drifting over the polynya region as it was

opening (Campbell et al., 2019), but these are the only in-situ data ever collected during one of its openings. Knowing of such upcoming re-opening, even if it is but a few days in advance, would allow for potentially re-routing autonomous sensors or nearby expeditions and hence obtaining precious data.

Polynyas are large openings in the winter sea ice, in both hemispheres, located by the coast or in the open ocean (Morales Maqueda et al., 2004; Smith Jr et al., 2007). By suddenly exposing the comparatively warm ocean to the cold winter atmosphere, they have a large impact on the entire climate system: they modify the whole water column (Gordon, 1978), contribute to deep water formation (Martin and Cavalieri, 1989) and hence impact the global ocean circulation (Heuzé et al., 2015a), and may be responsible for the observed warming of the deepest waters (Zanowski et al., 2015). Moreover, the vertical motion of water that they trigger ventilates the deep ocean and brings nutrients up, making polynyas a biological hotspot (Smith Jr et al., 2007). The largest of them all, the Weddell Polynya or Maud Rise Polynya, opens in austral winter in the Weddell Sea sector of the Southern Ocean. It was first observed at the beginning of the satellite era in winters 1974-1976, reaching a maximum area of $350\,000\ \mathrm{km}^2$ (Carsey, 1980). There was no large Weddell Polynya until 2016, but a "halo" of low sea ice concentration did regularly appear in the region (Lindsay et al., 2004; Smedsrud, 2005; de Steur et al., 2007), suggesting that the process that caused the polynya was still at play. The 1970s polynya has been extensively studied using models (e.g. Timmermann et al., 1999; Cheon et al., 2015; Cabré et al., 2017), but the exact, general-case opening process is still debated, owing to the lack of in-situ data. The many hypotheses fall into two broad categories:

– the atmospheric argument is based on relationships between the polynya and the strength or persistence of the Southern Annular Mode, the strength of the wind itself (e.g. Cheon et al., 2014; Francis et al., 2019; Campbell et al., 2019), or even on moisture transport (Francis et al., 2020);

– the oceanographic argument is two-fold: that the Maud Rise region is weakly stratified, hence prone to deep convection and polynya events (Kjellsson et al., 2015; Heuzé et al., 2015b; Wilson et al., 2019), and that the polynya opens when comparatively warm Circumpolar Deep Water is upwelled (Holland, 2001; Martin et al., 2013; Dufour et al., 2017; Cheon and Gordon, 2019).

Could the lack of in-situ data be compensated by satellite-based observations? The most common method to monitor sea ice globally consists in daily estimates of sea ice concentration from passive radiometers (Spreen et al., 2008), mostly in the microwave region (e.g. AMSR-E highest frequency of 89 GHz, or wavelength of approx. 3 mm). Passive microwave remote sensing is also commonly used for the determination of sea ice age (e.g. Maslanik et al., 2007), a proxy for its thickness and salinity. Passive microwave remote sensing products have a relatively low horizontal resolution of the order of 3 km, so (active) Synthetic Aperture Radar (SAR) has become common for high resolution applications such as lead (e.g. Murashkin et al., 2018) or melt pond detection (Mäkynen et al., 2014), sea ice drift tracking (e.g. Demchev et al., 2017), classification (e.g. Aldenhoff et al., 2018) and especially thickness retrievals (e.g. Zhang et al., 2016). To the best of our knowledge, none of these methods has been used for detecting that sea ice is about to open. We here use a subcategory of passive radiometers in the infrared region of the electromagnetic spectrum, known as the Advanced Very High Resolution Radiometer or AVHRR. AVHRR has been used to monitor sea ice since the late 1970s from multi-mission satellites, which gives it a higher-than-daily

coverage and makes it a quite robust method (Comiso, 1991). Our hypothesis, based on our preliminary study (Heuzé and Aldenhoff, 2018), is that the upwelling and/or heat loss to the atmosphere caused by sea ice thinning prior to the Weddell Polynya opening is detectable in these infrared data.

We here investigate whether spaceborne infrared imagery can be used to detect an upcoming re-opening of a polynya. In particular, we want to know whether these data can be used for pure binary detection of whether a polynya is upcoming (section 3), and also for determination of the process responsible (section 4). We base our study on the AVHRR Polar Pathfinder (APP), provided by the National Oceanographic and Atmospheric Administration (Key et al., 2019). After detailing our processing in section 2, in particular the cloud masking, we start by verifying how many polynya events have occurred since records began

(section 3.1) and their respective starting dates, using the Comiso "Bootstrap Sea Ice Concentrations from Nimbus-7 SMMR and DMSP SSM/I-SSMIS, Version 3.1" provided by the National Snow and Ice Data Center (Comiso, 2017). We determine criteria on the infrared brightness bands that successfully detect all these events two weeks ahead of the re-opening while returning no false positive (section 3.2). Finally, we evaluate whether the infrared time series can provide satisfactory proxies for lead opening or for upwelling, using in-situ and reanalysis data as reference (section 4).

## 2   Data and Methods

### 2.1   Data

In the interest of readability, we now indicate only the key characteristics of the data used for this study. The reader is encouraged to consult the corresponding data description papers that we cite. In this study, we first determine the dates of past polynya events using the daily product "Bootstrap Sea Ice Concentrations from Nimbus-7 SMMR and DMSP SSM/I-SSMIS,

Version 3.1", often referred to as "Comiso Bootstrap", provided at 25 km resolution by the National Snow and Ice Data Center, and available continuously since 1 November 1978 (Comiso, 2017, dataset doi: 10.5067/7Q8HCCWS4I0R).

    We then study these events using spaceborne infrared data, validated against in-situ hydrographic and atmospheric reanalysis data. Our region of interest (Fig. 1), hereafter referred to as "the polynya-prone region", is the fixed 778 000 $\mathrm{km}^2$ area that lies over the topographic feature Maud Rise, in the eastern Weddell Sea sector of the Southern Ocean (longitude 6° W to 12°

E; latitude 68° S to 60° S), where polynyas and halos have been consistently reported in the literature (e.g. Beckmann et al., 2001; de Steur et al., 2007; Campbell et al., 2019).

    The spaceborne infrared data come from the AVHRR Polar Pathfinder or APP, provided by the National Oceanographic and Atmospheric Administration (Key et al., 2019, dataset doi: 10.25921/X2X1-JR34). It provides twice-daily, 5 km gridded composites of all available AVHRR infrared brightness temperature data since 1982. We use only the ones acquired in the 6h

interval around 2 AM local solar time. The three bands that we use are commonly referred to as T3b (wavelength of 3740 nm), T4 (10800 nm) and T5 (12000 nm). Over the polynya-prone region, the APP data contain 90 000 grid cells.

    To determine whether specific infrared brightness temperature signals can be used as a proxy for either upwelling or lead opening, we compare APP to reanalysis and in-situ data. We use the hourly 2 m air temperature, 1000 hPa relative humidity, and 10 m horizontal wind components u and v from the European Centre for Medium-Range Weather Forecasts ERA5 hourly

reanalysis, provided on a 0.25° grid (dataset doi: 10.24381/cds.adbb2d47). Hydrographic data come from three moorings deployed along the Prime Meridian by the Alfred Wegener Institute, named AWI229 (Fahrbach and Rohardt, 2012a; Rohardt and Boebel, 2019, dataset doi: 10.1594/PANGAEA.793018 and 10.1594/PANGAEA.898781), AWI 230 (Fahrbach and Rohardt, 2012b, c, dataset doi: 10.1594/PANGAEA.793080 and 10.1594/PANGAEA.793082) and AWI 231 (Fahrbach and Rohardt, 2012d, dataset doi: 10.1594/PANGAEA.793089). Each mooring deployment has different characteristics, but they all measure temperature, salinity, pressure, and current velocity (not used here) at up to 15 irregularly spaced depth levels, every 15 min, for on average two years. The earliest deployments were in April 1996, and the program continues to date.

To determine the presence of leads, we use the position information from the autonomous Southern Ocean Carbon and Climate Observation and Modelling or SOCCOM (Riser et al., 2018) profiling float 5904468, obtained via the Southern Ocean Observing System (SOOS, https://www.soosmap.aq/platinfo/piroosdownload.aspx?platformid=9388). In particular, the float dataset features a position quality flag to indicate whether the position is correct (float at the ocean surface) or interpolated (float under ice). The float ascends every 10 days. If the median temperature over the depth range 20-50 m is above freezing during both the present ascent and the previous one, the float tries to surface; if the previous reading, 10 days prior, was at freezing temperature the float does not try to surface (personal communication Ethan Campbell and Dana Swift, 27 April 2021). That is, as the float requires two consecutive above-freezing profiles to surface, the float surfacing indicates for sure the presence of a lead on that date, and strongly suggests the presence of a lead 10 days prior.

## 2.2 Cloud masking of APP data

Clouds are a known issue for AVHRR data, especially in polar regions (e.g. Drinkwater, 1998). The first cloud filters adapted to the polar regions were designed by Yamanouchi et al. (1987), which imposed criteria on T4, T34 (T3b minus T4) and T45 (T4 minus T5) to detect thick, high and thin clouds, respectively. Saunders and Kriebel (1988) added a geographical/texture perspective, imposing criteria on 3 by 3 pixel areas, while Key and Barry (1989) added a temporal perspective, comparing each pixel from day to day. But these filters did not perform as well as expected, and we had to wait until Vincent et al. (2008) for extra criteria on T45 that can detect ice fog, and more recently Vincent (2018) to detect dust.

We aim to create a system that can work on individual images independently, so the approach of Key and Barry (1989) is not adapted. Likewise, Saunders and Kriebel (1988) is mostly based on day-time images, so not adapted to our work either. As detailed in Appendix A, after validation using the reference cloud mask MYD35_L2 v6.1 (Ackerman et al., 2017, dataset doi:10.5067/MODIS/MYD35_L2.006) over the period 4 July 2002 - 31 December 2018 that is common to both products, we chose to use the following three criteria to detect clouds (example on Fig. 1):

- T4 < 245 K (indigo, Yamanouchi et al., 1987);

- |T34| > 1.5 K (green, after Yamanouchi et al., 1987, see appendix A);

- T45 < 0 K or T45 > 2 K (yellow, Vincent et al., 2008; Vincent, 2018).

As shown on Fig. 1 and in Appendix A, these criteria are not perfect but they are powerful enough to detect most of the clouds. On average, only 14% of the cloud pixels are missed, but 15% of the pixels are also wrongly eliminated as cloudy even

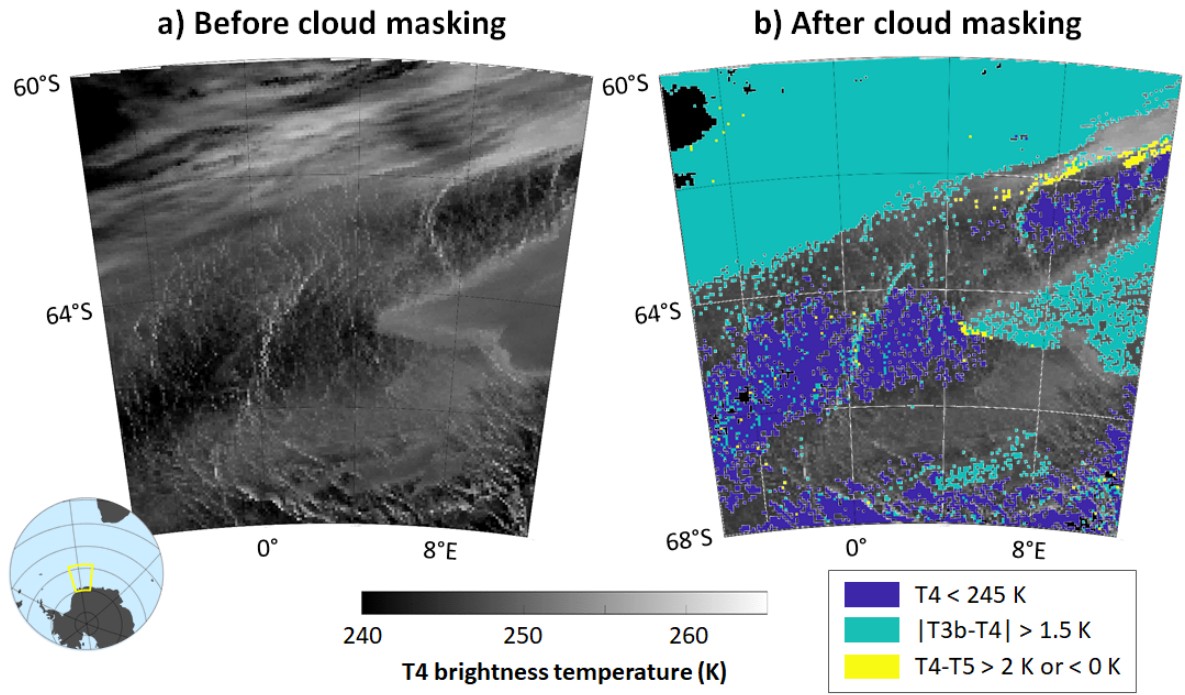

**Figure 1.** Infrared brightness temperature T4 from APP (provided by NOAA, Key et al., 2019) on 13 August 2009 before (a) and after (b) cloud masking. Colours indicate the different masking criteria on T4 (Yamanouchi et al., 1987, indigo), T34 (as detailed in appendix A, modified from Yamanouchi et al., 1987, green), and T45 (Vincent et al., 2008; Vincent, 2018, yellow). Insert indicates the location of the two images.

though they are clear. We strongly suspect that this is caused by the difference in acquisition time of the images used by each product, but could not verify this as APP is only directly available as a composite instead of distinct images. Moreover, leads

and polynyas generate a large heat and moisture flux (e.g. Cheon et al., 2014), so we need a cloud mask that is not so sensitive that it would mask the leads and polynyas and prevent us from detecting them. As our cloud masking is based on published literature and that cloud masking is not the topic of this paper, any further study of this question would be beyond the scope of this paper. Any pixel that meets any of the three criteria listed above is set to NaN in all three bands T3b, T4 and T5 for our calculations. The polynya events and the days leading to them that we study in this paper have a similar amount of cloud free

pixels to any of the non-polynya years in the series (supp. Fig. B1).

## 2.3 Methods

There are many criteria to detect a polynya, based on sea ice concentration or thickness thresholds (see Mohrmann et al., 2021, and references therein). In section 3.1, we detect polynyas by applying already published methods based on sea ice concentration threshold on each pixel and on the area-average. We show only the three different thresholds used by Gloersen

et al. (1992); Gordon et al. (2007); Campbell et al. (2019), but we tested and visually assessed, using the authors' experience, each option between 15 and 100% on the single-pixel sea ice concentration. We compute the polynya area by detecting the contour of the enclosed area with a sea ice concentration lower than the specific threshold (see codes if needed). An event is defined as an uninterrupted series of consecutive days with sea ice under that threshold (60% in this study); if there is a day with higher sea ice concentration in between, a new event is created. We will show in section 3.1 that there has been more than 20 polynya events over the last 40 years covered by the sea ice concentration product, and use the polynya events here detected to determine the characteristics of the infrared brightness temperature in the days leading to each event.

The overarching aim of our project is to eventually produce an automatic system that would be scanning the polynya-prone region, so we do not track individual polynyas and instead analyse the infrared brightness temperature time series over the whole region in section 3.2. We also computed daily anomalies of these infrared brightness temperatures relative to daily climatological values over the whole 38 year period covered by APP. For each day from 1982 to 2019, we produce time series of the geographical median, standard deviation, minimum and maximum infrared brightness temperature over the polynya-prone region for each band.

Finally, the atmospheric and hydrographic data used in section 4 are directly studied without further processing, except for the wind components. We produced for each polynya event a time series of what we hereafter call curl of the wind. We cannot compute the wind stress curl per se, as we lack the drag coefficient, which will anyway change depending on whether the polynya is open or close. So instead, we use a similar method as e.g. Petty et al. (2016) and work with the curl of the wind components u and v:

$$curl = \frac{\partial v}{\partial x} - \frac{\partial u}{\partial y}. \tag{1}$$

We are not studying the actual values of that curl, only whether it is positive (suggesting divergence that could open a lead) or negative (suggesting a downwelling, Marshall and Plumb, 2016).

## 3  Results: an infrared-based criterion valid for all polynya events

### 3.1  Polynya dates

We start by determining the dates of the polynya events that we want to further study according to three traditional criteria: sea ice concentration (SIC) minimum over the region lower than 15% (after e.g. Gloersen et al., 1992, black, Fig. 2) or lower than 60% (Campbell et al., 2019, blue), and average SIC lower than 92% (Gordon et al., 2007, red). As explained previously, we study the fixed "polynya-prone" region of longitudes 6°W to 12°E and latitudes 68°S to 60°S. We limit ourselves to the period 1st July to 31st October. As noted by Campbell et al. (2019) already, these methods return qualitatively similar results; it is only the number of days that differs for each criterion. They all agree there was some polynya activity in the late 1980s - early 1990s and in the early 2000s, referred to as the Maud Rise halo, already studied notably by de Steur et al. (2007). Then, the region was rather quiet until the widely reported 2016-2017 return of the polynya (Swart et al., 2018).

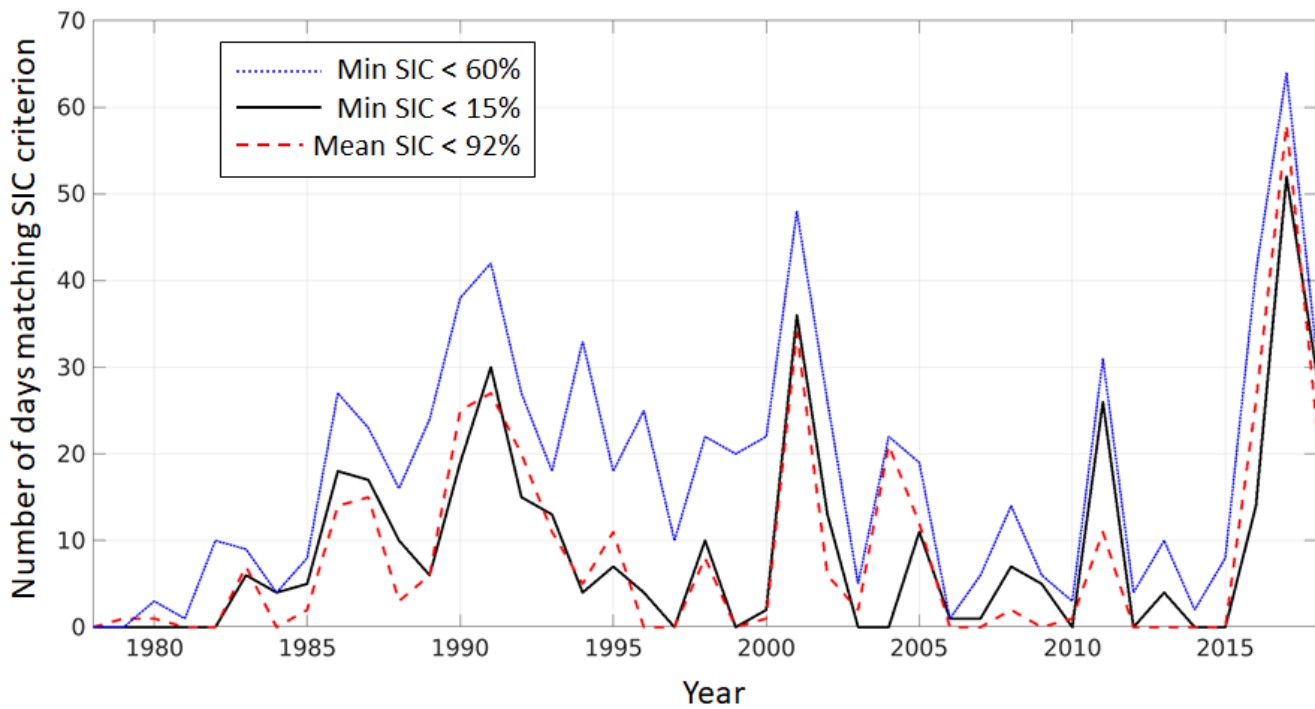

**Figure 2.** For each year of the sea ice time series (1978-2018), number of days between 1st July and 31st October with a polynya according to the three most common criteria in the literature: black, minimum sea ice concentration (SIC) lower than 15% (e.g. Gloersen et al., 1992); blue, minimum sea ice concentration lower than 60% (Campbell et al., 2019); red, average sea ice concentration over the (fixed) polynya-prone region lower than 92% (after Gordon et al., 2007).

The 60% criterion represented by the blue line probably includes late freeze-up or early melt events. The actual method used by Campbell et al. (2019) does not use fixed dates for winter, but instead limits itself to the period that starts one week after the first 90% SIC and finishes one week before the last 90%. For consistency with the other two methods, we used fixed winter dates for all criteria. We then visually checked the individual images to separate such late freeze-up / early melt from the actual
polynya events. This confirmed that although most sensitive, the 60% criterion used by Campbell et al. (2019) performs best (not shown).

The characteristics of the events thus detected are given in appendix table B1. We have 24 events over 11 years, which yields 30 polynyas because 5 events have two to four polynyas in the region simultaneously. Note that all durations make the polynya disappear on 1st November because of our end date criterion.
These 30 polynyas open at key locations in the region (Fig. 3). The maximum number of polynyas opening at the same grid cell, nine, is on the north-east flank of Maud Rise, with most of the others opening over Maud Rise or on its south-west flank. The central role of Maud Rise as shown by Holland (2001) is obvious. There is also a non-negligible number of grid cells in

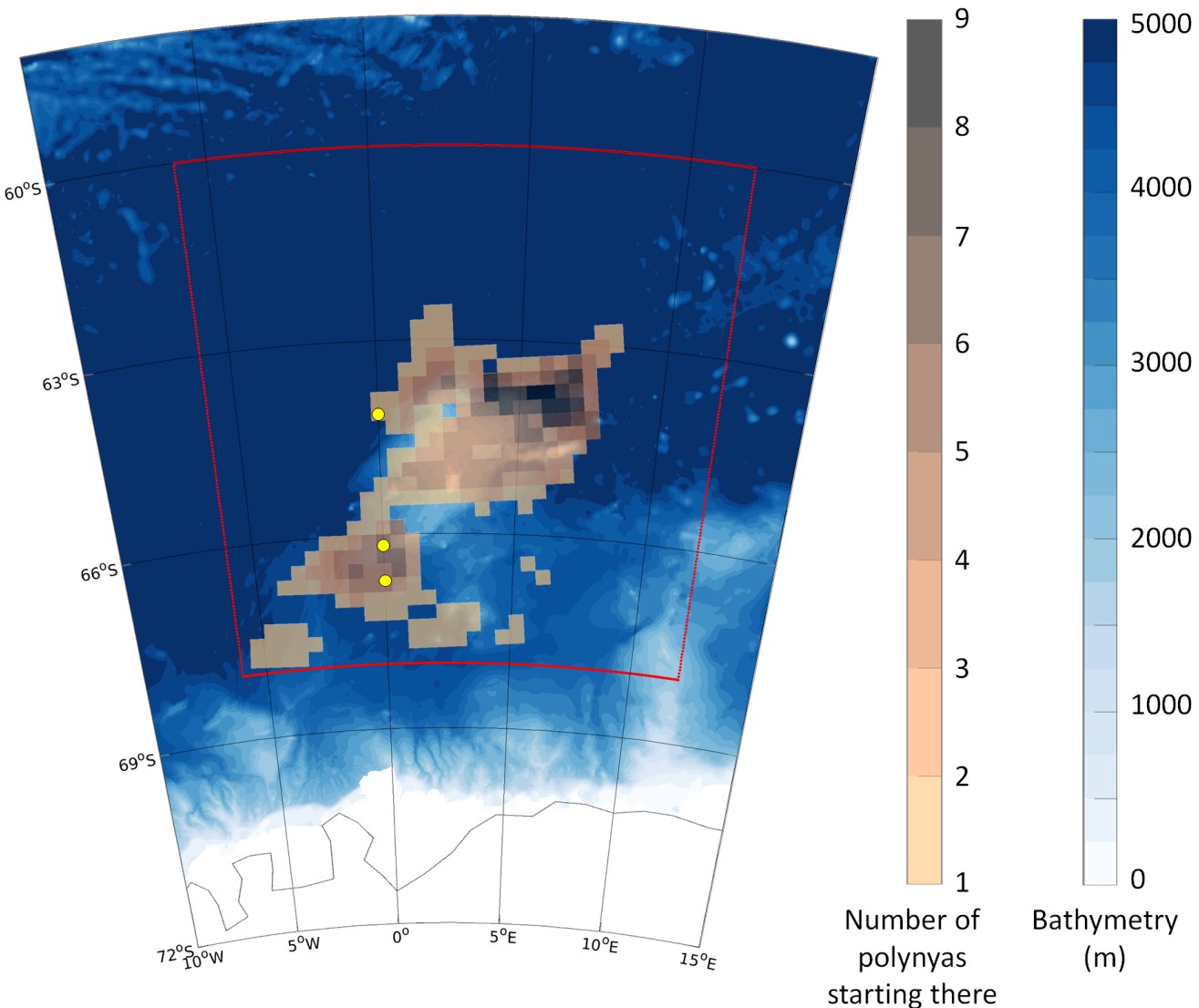

**Figure 3.** Out of the 30 polynyas detected, how many start at each location (transparent shading). Background: bathymetry of the region from GEBCO Compilation Group (2019). Red contours indicate the polynya-prone region studied in this manuscript. Yellow dots mark the location of the three moorings used in section 4: AWI229 (north), AWI230 (middle) and AWI231 (south).

the southern part of our region with at least one opening. Unlike in models (e.g. Martin et al., 2013), Fig. 3 shows no opening over the open ocean.

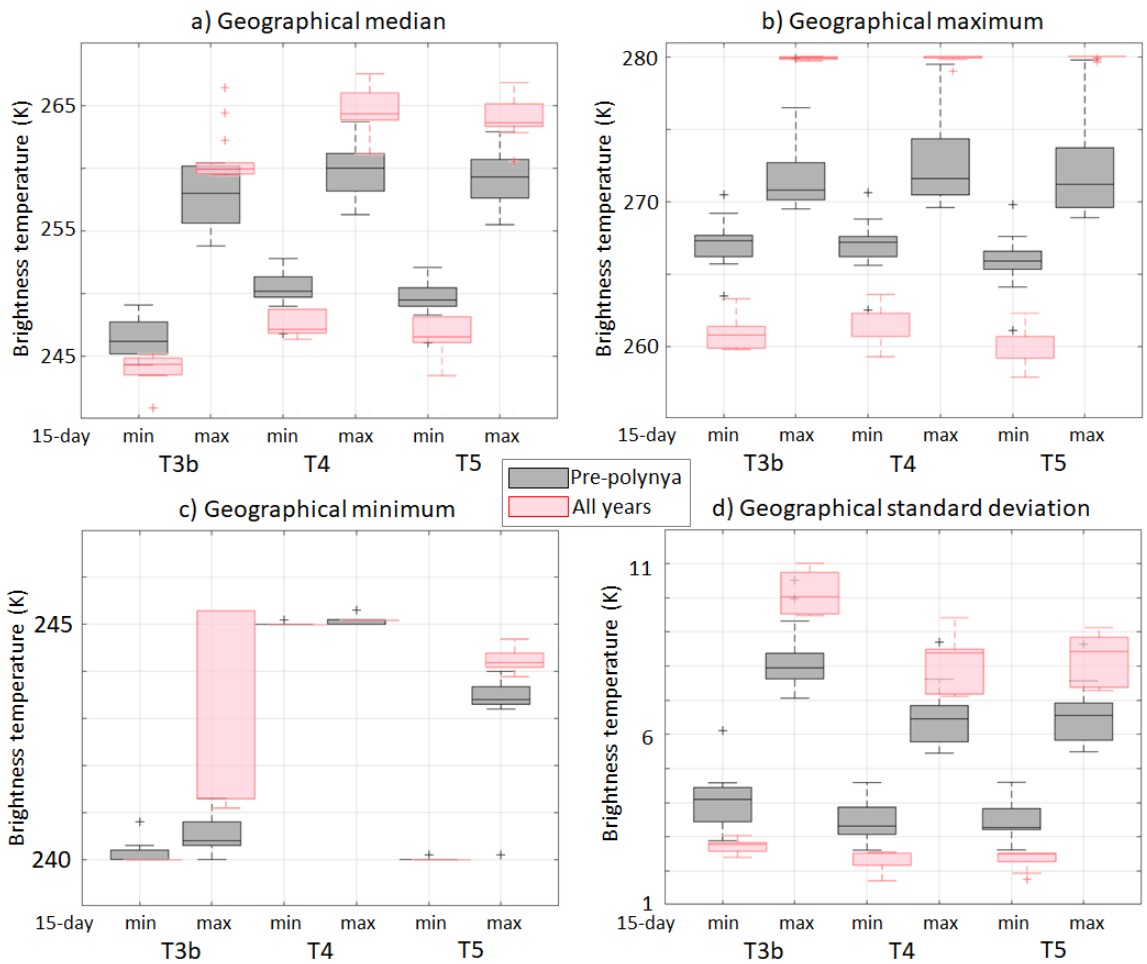

**Figure 4.** Over the polynya-prone region, for all bands, whisker plot of the minimum (x-axis) and maximum (y-axis) infrared brightness temperature over the time period from 15 days before the polynya opens until the day of opening of the geographical median (a), maximum (b), minimum (c) and standard deviation (d). Grey colours show the values of the years where a polynya opens; pink, for the same dates in all years.

## 3.2 Infrared-based early detection criteria

In the previous section, we determined the dates of 24 polynya events (giving 30 actual polynyas) from sea ice data dating back to 1978. We now investigate, in the timeseries of infrared brightness temperature from APP, whether all these events share something in common, especially in the 15 days leading up to the event. We present the 30 days prior to the events in section 4 but we found that for the current purpose, 15 days are enough. As explained in the Methods section, this "something in common" needs to be easily detectable by a crude automatised system, hence we computed basic single-image properties

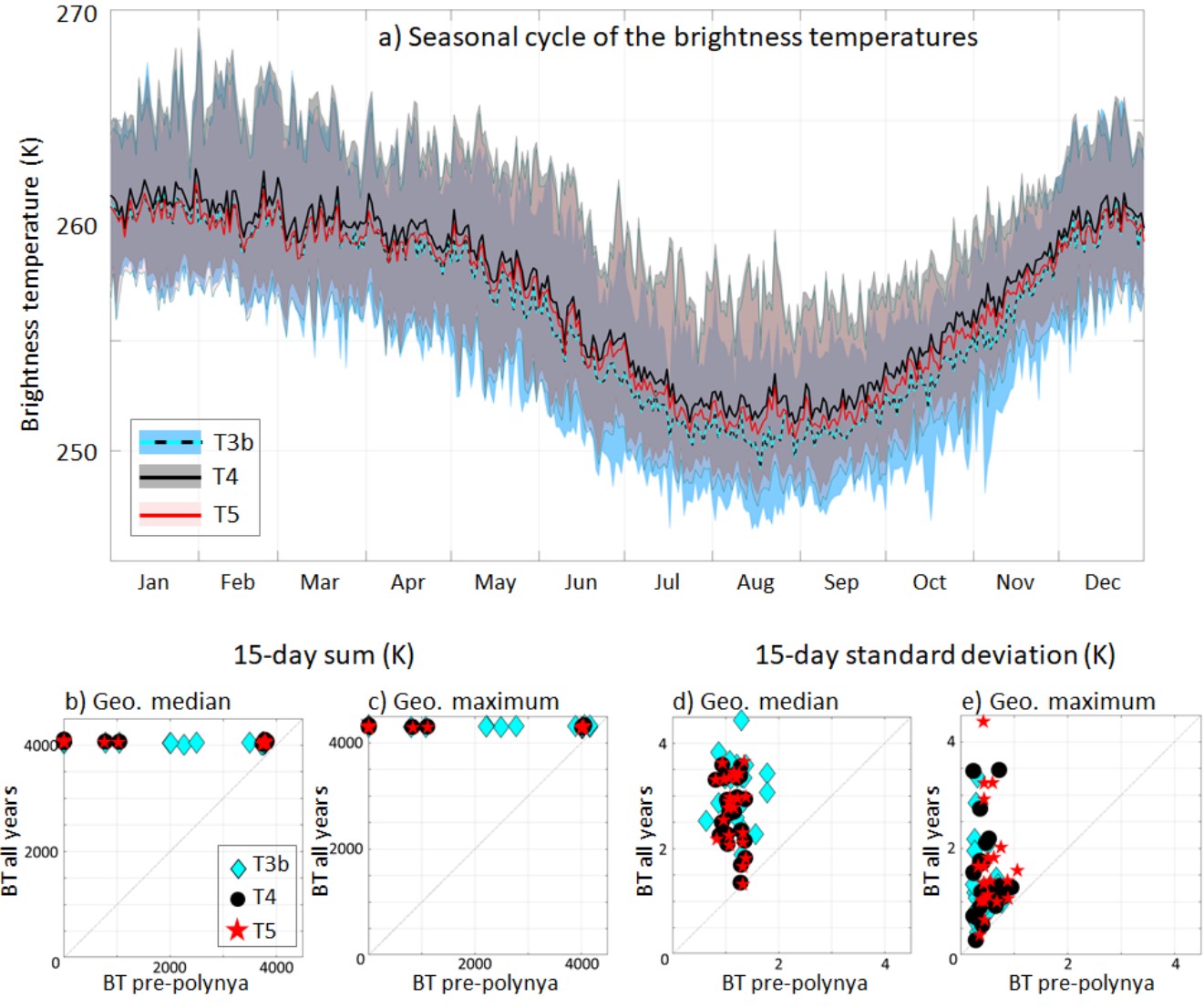

**Figure 5.** For the three bands, a) 1982-2019 median (thick line) and 10 to 90 percentiles (shading) infrared brightness temperature value for each day of year. b) to d) demonstrate the narrow range of the infrared brightness temperature (BT) statistics over the 15 days leading to each polynya (x-axis) compared to that on the same date in all years (y-axis): 15-day sum of the geographical b) median and c) maximum; 15-day standard deviation of the geographical d) median and e) maximum. One point per polynya event: cyan diamonds for the temperature band T3b; black circle for T4; red star for T5. Thin black line is the unit line.

over the entire polynya-prone region: the geographical median, minimum, maximum and standard deviation. We analyse the range of their values in the 15 days leading to the event and compare them to years with no polynyas (Fig. 4).

We want to find a criterion that would not only robustly detect a polynya, but also not flag any false positive. Fig. 4 shows that finding a simple threshold criterion will not be possible. For all geographical statistics and all bands, the range of the polynya values (grey on Fig. 4) is contained in the range of the years with no polynya (pink): the 15-day minima are larger when there is a polynya; the maxima, lower. We find the same results when looking at T34, T45 and T35, as well as all bands' anomalies (not shown).

One reason for the impossibility of finding an absolute threshold may be the seasonal cycle both in infrared brightness temperature and in anomalies (Fig. 5a). Polynyas are detected any time between early July and late October (supp Table B1). It is thus not surprising that a threshold that would successfully detect a polynya in August, when the brightness temperature is at its minimum but anomalies in T4 and T5 are largest, would fail in October as the temperature is higher and anomalies lower (Fig. 5a).

Instead, the narrower range of values during a polynya shown on Fig. 4 suggests that a criterion based on the persistance of values may be more adapted. Indeed, we find that years with polynyas have systematically lower 15-day sum or 15-day standard deviation than their non-polynya counterparts, in any band, with any geographical statistics although it is clearest with the geographical median and maximum (Fig 5 b to d). We find the same result when using the overall climatology instead of only the years with no polynya. We find no false positive.

In summary, a polynya is going to open if the 15-day sum or 15-day standard deviation of its geographical median or maximum over the polynya-prone region is lower than the 15-day sum or standard deviation produced using the climatology at the same date. See also Fig. 6 for a summary in flowchart form. Such low variability could be atmospheric driven, the result of a blocking-event. Studying the large scale atmosphere dynamics is beyond the scope of this paper, and a brief comparison of the 2-m air temperature over the polynya-prone region in the 15 days leading to each event (supp. Fig. B2) shows a lot of variability in the atmosphere. That is, pronounced changes in temperature for each event (e.g. up to 25° increase in 2 days for the first event), and no consistency among the events in magnitude and sign of the variations. The blocking explanation seems unlikely and would anyway be in opposition to the latest research on the atmospheric drivers of the Weddell polynya (see e.g. Francis et al., 2019, and references therein). It is thus more likely that the low-variability in infrared brightness temperature is the result of latent heat exchanges in the ocean-ice-atmosphere system as sea ice melts and refreezes.

We have determined a method based on infrared brightness temperature data to detect a polynya before it opens. Can we extract more information out of these infrared brightness temperatures? For example, information about the process that is responsible for opening the polynya. This is what we determine in the next section.

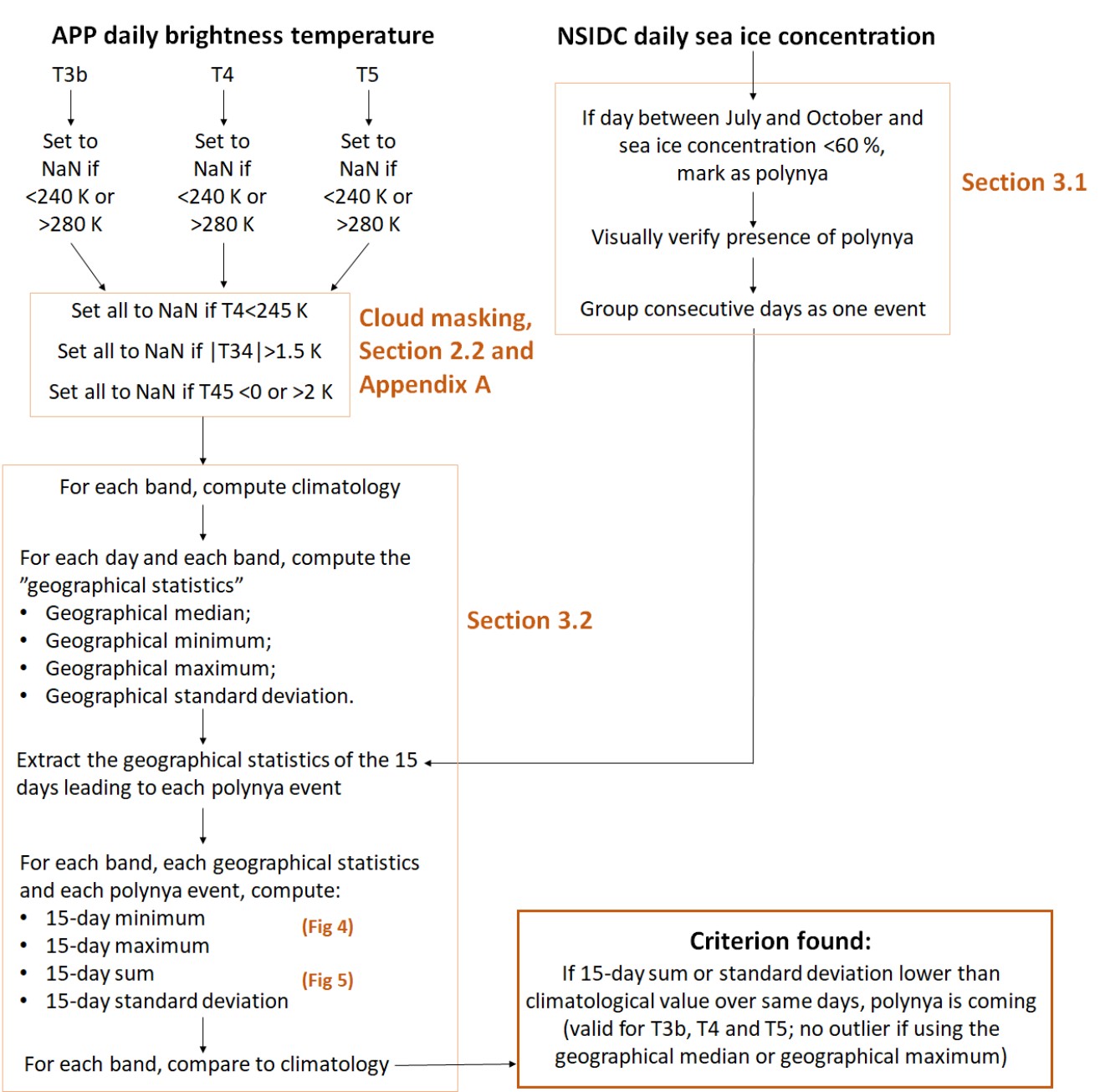

**Figure 6.** Flow chart summarising our methods and the findings of section 3.2.

## 4 Can infrared brightness temperature indicate why the polynyas open?

There are two ways to open a polynya (Morales Maqueda et al., 2004):

- sensible heat / open ocean polynyas form when the sea ice is melted from below by the ocean. It requires that a comparatively warm water mass is upwelled;

- latent heat / coastal polynyas in contrast form when the sea ice is pushed away by the wind, often in response to wind divergence.

Past literature suggests that both ways may be detectable in the brightness temperature series. Heuzé and Aldenhoff (2018) hypothesised that oscillations in the infrared brightness temperature, especially in T4, in the days before the polynya opens might reflect oceanic convective movements. Their argument is that as the warm water is being upwelled, more heat is going through the ice. Leads could also be detectable using the difference between bands T4 (most adapted to ice) and T5 (most adapted to open ocean Vincent et al., 2008) or T45, where a decrease in T45 would indicate a lead. We first investigate the hypothesis that T4 might be used as a proxy for upwelling of the comparatively warm Circumpolar Deep Water (CDW).

### 4.1 Are oscillations in T4 the result of upwelling?

The warm water mass that is upwelled when a polynya opens in the Weddell Sea is the Circumpolar Deep Water (e.g. Dufour et al., 2017, and references therein). According to the literature, there are at least three non-independent ways through which upwelling of warm and salty CDW can affect the sea ice and that would have an effect on the brightness temperature:

- A thin enough ice may open leads in response to the upwelling (e.g Campbell et al., 2019), resulting in a heat loss from the ocean to the atmosphere.

- Increased ocean surface temperature melts the sea ice bottom, reducing ice thickness, and/or results in increased conduction of heat through the ice (e.g. McPhee and Untersteiner, 1982); we here ignore the enhancing effect of momentum flux (McPhee, 1992), as we have no information about the ice bottom topography.

- Increased ocean surface salinity results in increased convection through the ice brine channels (e.g. Lytle and Ackley, 1996). Also, this increase in salinity lowers the freezing temperature, further enhancing the second effect.

All three processes result in heat and moisture loss from the ocean, which is what we hypothesise could be visible in the AVHRR data as an increase in brightness temperature T4. We test this hypothesis in this section.

Along the Prime Meridian, CDW's temperature maximum is located around 300 m depth (red shading on Fig. 7) and salinity maximum around 400 m (Fig. B3). On the three moorings, all hydrographic sensors are in the CDW (symbols on Fig. 7 and supplementary Table B2). The shallowest ones are above the core, that is, more than 100 m shallower than the temperature and salinity maxima, and can therefore be used to observe potential upwelling.

For hydrographic sensors above 200 m, we find a positive and significant correlation between the daily-average mooring temperature recorded by that sensor and the T4 brightness temperature difference at the same location over the period 1st July

to 31st October (depth of sensor and correlation values in supplementary Table B2). Moreover for all years, the correlation increases with depth until it reaches a maximum between 200 and 700 m (not shown). It is worth noting that over the same 1st July to 31st October period, the vertical displacement of the sensors does not exceed 20 m, i.e. the temperature change is not

caused by a displacement of the sensor but by a displacement of the warm water. When salinity data are available, we also find positive correlations between T4 and the salinity measured by the mooring (supplementary Table B2), which further indicates that variations in T4 mirror variations in warm and salty CDW. The T4 data are too patchy to robustly determine the increase in brightness temperature that corresponds to upwelling; of the eight polynya opening dates during which the moorings were deployed, the longest non-interrupted T4 series at the mooring location reach only 4 days before the opening, on two occasions

(July 1999 and October 2004). Both moorings 229 and 230 recorded a decrease in temperature in October 2004, but an increase of 0.6 K in 1999. That increase in the mooring temperature corresponded to an increase in T4 of 18 K, significantly larger than the standard deviation of T4 over the previous 30 days (4.6 K).

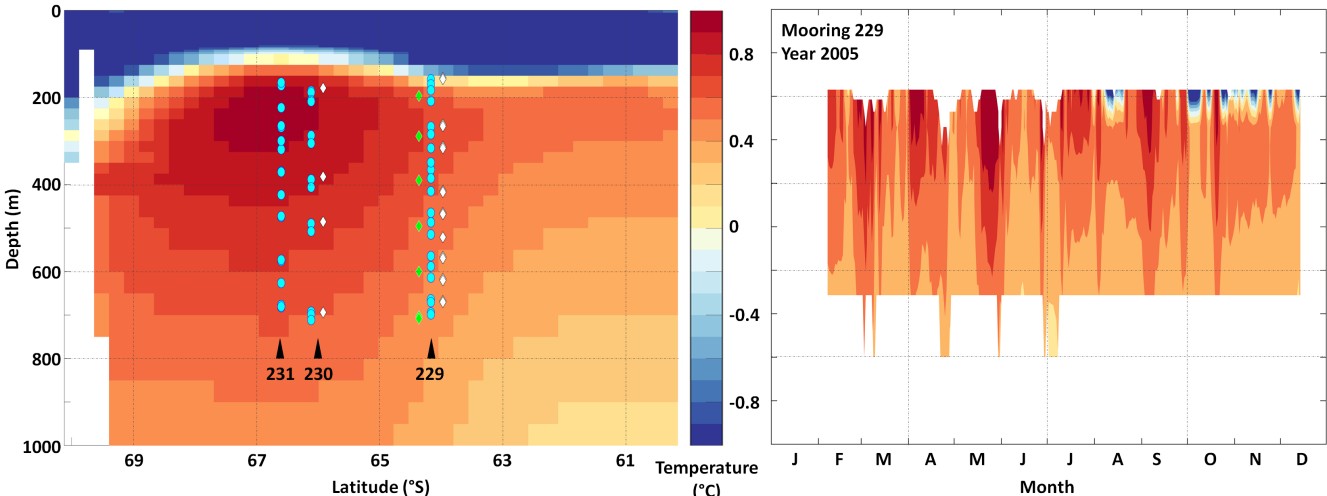

**Figure 7.** Left: Climatological temperature along the Prime Meridian, from Locarnini et al. (2018), showing the comparatively warm Circumpolar Deep Water in red. For each mooring line (229, 230, and 231), the symbols indicate the depth of the mooring temperature sensors during deployments that coincided with polynya events: white diamonds for 1999 and 2000; cyan circles for 2004 and 2005; and green diamonds for 2016. The latitude shift between deployments of the same mooring is for readability only. Salinity section available as supplementary figure B3. Right: on the same vertical axis (depth from 0 to 1000 m), time series of daily median temperature as recorded by mooring 229 in 2005.

Are changes in ocean temperature really what we see in the variations in T4? There are other factors that can impact the brightness temperature retrievals, such as changes in cloud cover, air temperature, humidity, wind speed, and even sea

ice emissivity and scattering (e.g. Bushuev et al., 2007). As cloud cover has been addressed with the cloud masking (see earlier sections), and sea ice surface properties are dependent on air temperature and wind speed, we here concentrate on air temperature, relative humidity and wind speed at the surface (Table 1, atmospheric parameters from ERA5 reanalysis, see

Methods). We expect positive correlations between T4 and both ocean and air temperatures, negative correlation between T4 and the relative humidity, and correlations in either direction with the wind speed depending on whether it leads to an upwelling, to surface cooling, or changes surface scattering. Considering only the two weeks leading to a polynya opening (unlike in the previous paragraph, where we considered the whole winter), we find that in 1999, 2000, 2004 and 2005, the strongest correlations tend to be between T4 and the mooring temperature, albeit with some variability depending on the mooring. For 2016 and 2017 where only mooring 229 can be used, the air temperature dominates. This suggests that in the first four years, changes in T4 may be an acceptable proxy for upwelling as we had hypothesised, whereas in 2016 and 2017, we cannot conclude with certainty that T4 indicates that a polynya is triggered by upwelling.

**Table 1.** For the polynya events since the first mooring deployment in 1996, correlation between the brightness temperature T4 and the ocean temperature from the mooring's shallowest sensor $\theta_O$, the 2 m air temperature $T_A$, the 1000 hPa relative humidity $r$, and the 10 m wind speed $V$ from ERA5 (see Methods), at the location of the thee moorings 229 to 231, over the 15 days before the polynya opens. For the mooring temperature, a lag of up to 10 days is permitted in order to account for the upwelling velocity. N/A indicates the absence of sensor shallower than 300 m. Only significant correlations (at 95%) are indicated.

| Polynya | Mooring 229 | | | | Mooring 230 | | | | Mooring 231 | | | |
|---|---|---|---|---|---|---|---|---|---|---|---|---|
| | $\theta_O$ | $T_A$ | $r$ | $V$ | $\theta_O$ | $T_A$ | $r$ | $V$ | $\theta_O$ | $T_A$ | $r$ | $V$ |
| 1999 Jul | 0.44 | - | 0.26 | -0.38 | 0.61 | 0.91 | - | 0.34 | N/A | 0.82 | - | 0.61 |
| 1999 Oct | 0.70 | 0.67 | - | - | 0.94 | 0.38 | - | - | N/A | - | - | 0.54 |
| 2000 Jul | 0.60 | 0.73 | -0.87 | - | 0.74 | 0.39 | - | -0.70 | N/A | - | 0.30 | - |
| 2000 Oct | 0.73 | 0.88 | 0.82 | 0.43 | 0.42 | -0.83 | -0.64 | -0.75 | N/A | 0.64 | - | - |
| 2004 | - | - | -0.28 | - | 0.46 | - | - | - | 0.78 | -0.32 | - | -0.33 |
| 2005 | 0.89 | 0.64 | - | - | 0.66 | 0.55 | 0.66 | 0.27 | - | 0.64 | 0.33 | - |
| 2016 | 0.50 | 0.94 | - | 0.54 | N/A | 0.45 | 0.42 | - | N/A | 0.58 | - | 0.61 |
| 2017 | 0.23 | 0.48 | 0.30 | - | N/A | - | - | - | N/A | 0.76 | - | 0.53 |

Are these variations in CDW upwelling in its strict meaning? As reviewed by Campbell et al. (2019), upwelling did occur in the years the polynya opened. However, wind-driven upwelling is a comparatively slow process, with typical vertical velocities between 0.1 to 4 m/day, up to 20 m/day for wind speeds of 20 m/s (Campbell et al., 2019; Häkkinen, 1986, respectively). In the polynya years, we detected in section 3.1, daily maximum wind speeds at the mooring location vary between 16 m/s in 2004 and 25 m/s in 2016. For the signal to travel from the shallowest mooring at 150 to 200 m depth to the near-surface, i.e. to be at the surface measured by T4, would hence require at least 8 days. Yet, allowing for a lag of up to 15 days, we find that the correlation between T4 and each mooring's temperature over July-October is maximum for 1-5 day lags (values not shown; see also Fig. 7, right). It is more likely that the water column is moved upward by a faster process such as eddies, which have been suggested as the main trigger of Weddell Polynyas (Holland, 2001). Moreover, the correlation between daily wind curl at the location of the mooring and the daily temperature of the shallowest mooring sensor is rare, and when it does occur, it is negative. For the polynya years in particular, this correlation is -0.15 (90% significance) for mooring 229 in 1999, and

-0.22 and -0.24 (both 95% significance) for moorings 229 and 230 respectively in 2004. No significant correlation is found in 2000, 2005, 2016 or 2017. A negative correlation indicates that a positive curl of the wind, i.e. upwelling-favourable, is in fact associated with a cooling of the upper water column. That is, changes in the water-column may be the result of an opening of the ice, rather than wind-induced upwelling.

In conclusion, there is a positive correlation between oscillations in infrared brightness temperature T4 and temperature and salinity in the CDW layer, i.e. vertical displacement of the CDW is associated with changes in T4. For 2016 and 2017, other processes seem to have a larger influence on T4 retrievals, so we would not recommend using T4 as a proxy for upwelling without ancillary in-situ data. In strict terms, this displacement is most likely the result of eddies rather than slower upwelling, or even lead-induced cooling (as suggested by the negative correlation with the curl of the wind). We now investigate whether these ice openings can be detected, using the infrared brightness temperature difference T45.

## 4.2 Decrease in T45 as a proxy for lead opening

The brightness temperature difference T45 is higher above ice or snow than above water (Vincent et al., 2008). Consequently, we here hypothesise that at a given location, a decrease in T45 can be indicative of an ice opening, most likely by leads. We first test this hypothesis by computing the correlation between T45 and the curl of the wind that we just discussed in the previous subsection. Lead opening corresponds to diverging winds, i.e. a positive curl, so we expect a negative correlation between T45 and the curl of the wind. Careful examination of high resolution products, e.g. Synthetic Aperture Radar, shows that the region often features leads, even in non-polynya years (not shown). Therefore, we perform the correlation over the whole winter 1st July - 31st October period, for all years. We do find significant (95%) negative correlations throughout the study area when comparing the curl of the wind to T45 at the same location over July-October for all years (not shown), with values as low as -0.24. When selecting only the years with a polynya (Fig. 8a), the correlation values reach -0.49. That is, for all winters over 1982 to 2018, increases in curl of the wind are associated with decreases in T45, and this relationship is clearer when selecting the winters where a polynya opened. In both cases, the correlation map is very patchy, most likely because of the large number of cloudy days (see Fig. 8b as an example).

This correlation is no proof that a lead had indeed opened when T45 decreased, only that the atmosphere was favourable to lead opening. We therefore now turn to the only direct evidence of lead opening that we could find: the surfacing of the autonomous floats used by Campbell et al. (2019). The reader will find more information about those floats in Riser et al. (2018). The one parameter we are interested in is the position flag, which indicates whether the float location is certain, i.e. really at the surface of the ocean with its antenna in the air, or interpolated, i.e. under ice. A float at the surface means that there was a lead at that location for the float to surface through on that day and 10 days before (see section 2.1). Looking at the period 1st July to 31st October only, no float surfaced in 2014 (deployment year), 2015 or 2016, but float 5904468 did surface in 2017, on 13th August at 65.0810°S, 0.0630°E, then on 13th September at 65.1560°S, 0.1930°W, i.e. 14 km away. As the float ascends every 10 days and surfaces only if ocean temperatures were above freezing during the previous ascent, this means that:

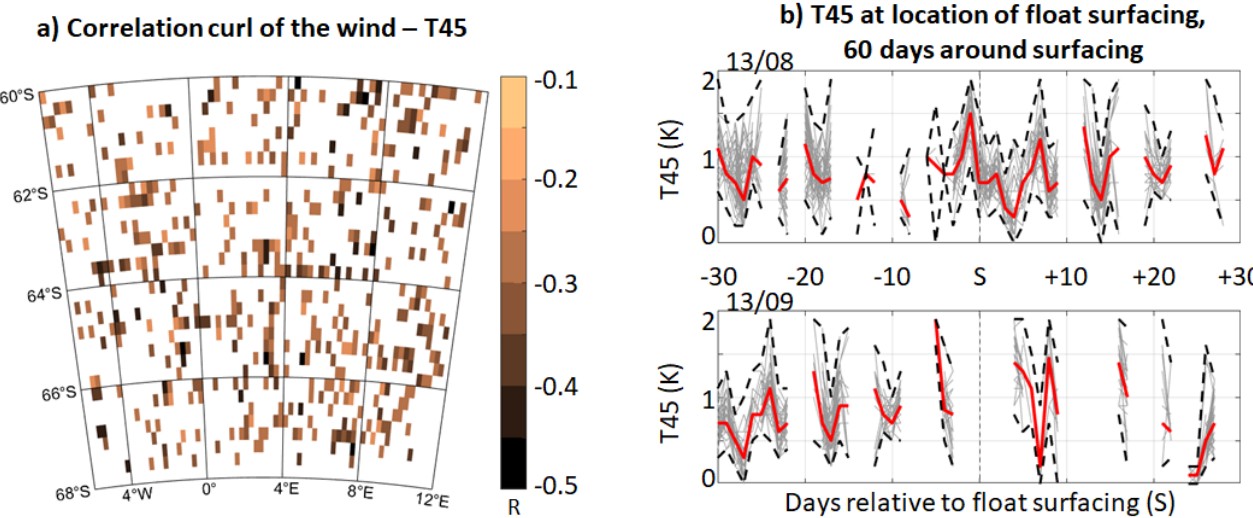

**Figure 8.** a) Correlation between daily curl of the wind and T45 for the polynya years over 1st July to 31st October, with T45 interpolated onto the wind grid. Only significant negative correlations are shown. b) Time series of all T45 (grey) in a 14-km radius around the location of the float surfacing, with median (red) and extrema (black dashed) highlighted, 30 days before to 30 days after surfacing on 13 August 2017 (top) and 13 September 2017 (bottom).

  – 24th July 2017 was ice-covered;

  – 3rd August 2017 was above freezing, most likely ice-free;

  – 13th August 2017 was confirmed as ice-free by surfacing;

  – 24th August 2017 the float was presumed trapped under ice (Ethan Campbell and Dana Swift, personal communication, 27 April 2021);

  – 3rd September 2017 was above freezing, most likely ice-free;

  – 13th September 2017 was confirmed as ice-free by surfacing.

For the two surfacings on 13th August and 13th September, we determine whether T45 noticeably decreased around the surfacing time and/or was noticeably lower after the surfacing compared to before (Fig. 8b). Unfortunately, the T45 data is too patchy to know. Looking up to 30 days before and after, i.e. accounting for three surfacing attempts either side of the surfacing

325 day, and even by selecting points in a 14 km radius around the surfacing position to account for the float drift, we lack data for most days, in particular for "ascent" days (every 10 days, vertical bars). We can say that on 13th August (Fig. 8b, top), T45 dropped compared to its value the day before (median, from 1.5 to 0.7 K). The T45 value on surfacing day (0.7 K) is lower than it was 20 (1.2 K) and 30 days (1.0 K) before, all days where the float ascended but did not surface, and somewhat similar

to that of 20 days after (0.8 K), when the float measured temperatures above freezing again but was not allowed to surface yet. We have no data 10 days before, when the ocean subsurface was first detected above freezing and consequently the float was not allowed to surface. We have no T45 data either when the float surfaced again on 13th September (Fig. 8b, bottom). We can only see what we just discussed: 30 days before, i.e. 13th August surfacing, the median T45 over that region is low (0.6 K), and so is it on 3rd September (0.7 K) when the float was not allowed to surface but the area was presumed ice-free.

In summary, T45 is significantly, negatively correlated with the curl of the wind, and we do see a decrease in T45 to values below 1K before the SOCCOM float surfaced in leads in 2017. These two results are however not conclusive as they are strongly impedded by APP's sensitivity to clouds. Besides, the leads that we tried to detect are too narrow compared to APP's horizontal resolution. We recommend instead the use of products that are not affected by clouds and are higher resolution (e.g. Zwally et al., 2008; Murashkin et al., 2018) and/or to try and detect only large leads (e.g. Bröhan and Kaleschke, 2014; Reiser et al., 2020).

## 5   Conclusions

The first aim of this paper was to determine criteria on 38 years of spaceborne infrared imagery (APP) to detect an upcoming reopening of the Weddell Polynya. Using the Comiso Bootstrap sea ice concentration, we generated a time series of past polynya events over Maud Rise and obtained 24 events since 1980, or 30 polynyas as some days had several polynyas opened simultaneously (Fig 2). The widely accepted narrative is that there had been no polynya in the Weddell Sea since "the" Weddell Polynya of 1974-1976 when the polynya unexpectedly re-opened in 2016 (e.g. Swart et al., 2018). Yet, our study is but one of many that found once again that there has in fact been many polynyas / halos in the region in the forty years in between (e.g. Lindsay et al., 2004; Smedsrud, 2005; de Steur et al., 2007; Campbell et al., 2019).

Although no absolute infrared brightness temperature threshold criterion could be found, all geographical statistics exhibit a reduced variability in the two weeks leading to any of these events compared to the same date in the rest of the dataset and in the climatology. The 15-day sum and 15-day standard deviation, in particular of the geographical median and maximum, successfully detected the events without finding false positives. The next step would be to check whether these criteria are still valid for other Antarctic open ocean and coastal polynyas (e.g. the Amundsen Sea Polynya, Randall-Goodwin et al., 2015), and also Arctic polynyas (e.g. North Water and North Greenland polynyas, Preußer et al., 2015; Ludwig et al., 2019).

Finally, we investigated whether spaceborne infrared data could be used not only to detect that the polynya is about to re-open but also which process causes this re-opening. The two processes we investigated here are warm water upwelling and deep convection (e.g. Holland, 2001; Martin et al., 2013; Dufour et al., 2017; Cheon and Gordon, 2019), and wind-driven lead opening (e.g. Gordon et al., 2007; Cheon et al., 2014; Campbell et al., 2019; Francis et al., 2019). Comparing the infrared data to atmospheric reanalysis and mooring hydrographic time series, we found that oscillations in T4 may be used as a proxy for vertical movements of warm water, although caution should be exercised as other processes may occasionally dominate the retrieval. Comparing the infrared data to atmospheric reanalysis and autonomous profiler positions, we find that a decrease in T45 may be used as a proxy for lead opening, but the lack of data because of clouds and the relatively coarse resolution of

APP compared to the leads does not make APP the most relevant product for this purpose. On the other hand, as spaceborne infrared is very sensitive to clouds, it has long been used to study clouds themselves (e.g. Key and Barry, 1989; Pavolonis and Key, 2003; Wang and Key, 2005). APP could then become the most relevant product to study atmospheric rivers, which have
365 recently been suggested as another crucial process leading to the 1973 and 2017 Weddell Polynya (Francis et al., 2020).

*Code availability.* Codes available at https://github.com/cheuze/Polynya_EGU_TC

*Data availability.* Sea ice data, dataset doi:10.5067/7Q8HCCWS4I0R, freely available online at https://nsidc.org/data/nsidc-0079.
APP infrared data, dataset doi:10.25921/X2X1-JR34, freely available at https://www.ncei.noaa.gov/data/avhrr-polar-pathfinder/access.
ERA5 data, dataset doi:10.24381/cds.adbb2d47 and 10.24381/cds.bd0915c6, freely available at https://cds.climate.copernicus.eu/cdsapp/
dataset/reanalysis-era5-single-levels and https://cds.climate.copernicus.eu/cdsapp/dataset/reanalysis-era5-pressure-levels respectively.
Mooring data freely available on PANGAEA at https://doi.pangaea.de/10.1594/PANGAEA.[dataset numbers in reference list].
SOCCOM profiler data freely available via SOOS at https://www.soosmap.aq/platinfo/piroosdownload.aspx?platformid=9388.

## Appendix A: Validation of the cloud masking

In section 2.2, we detect cloud pixels using the infrared brightness temperature thresholds of Yamanouchi et al. (1987), Vincent et al. (2008), and Vincent (2018). We here verify their methods using the Level 2 MODIS Cloud Mask product MYD35_L2 v6.1 (Ackerman et al., 2017, product DOI:10.5067/MODIS/MYD35_L2.006), obtained from https://search.earthdata.nasa.gov/ (last accessed 5 January 2021).

We obtained all MYD35_L2 granules available from 4 July 2002, when the product was first available, to 31 December 2018, over the region in the latitude range 68°S to 60°S and in the longitude range 6°W to 12°E, totalling nearly 65 000 granules. Latitudes and longitudes were available from the matching geolocation files (MYD03). For comparison with the APP data analysed here, we kept only the granules that were collected between three hours before and three hours after 2 am local solar time. In that time interval, we kept only the granules where the cloud mask was determined (first bit field equal to 1, Strabala, 2005), resulting in a number of granules used here varying between one and three for each day. For each 1 km by 1 km pixel in each such granule, that pixel is cloudy if both bit fields 2 and 3 are equal to 0, and clear otherwise (Strabala, 2005). A further test (not shown) where both "cloudy" and "uncertain clear" (bits 2 and 3 set to 00 and 01 respectively) were treated as cloudy yielded no significant difference. Finally, we interpolated the MYD35_L2 cloud mask onto the 5 km grid used by APP.

For APP, we consider that a pixel is clear if it matches all three criteria:

- T4 $\geqslant$ 245 K (Yamanouchi et al., 1987);

- $|T34| \leqslant$ 2 K (Yamanouchi et al., 1987);

- and 0 K $\leqslant$ T45 $\leqslant$ 2 K (Vincent, 2018; Vincent et al., 2008).

For all 90 000 pixels of each daily file, we determine where:

- both MYD35_L2 and APP detect no cloud (cyan on supp. Fig. A1d);

- both MYD35_L2 and APP detect a cloud (grey);

- MYD35_L2 detects no cloud but APP does (orange);

- MYD35_L2 detects a cloud but APP does not (black).

The case where MYD35_L2, used as reference here, detects a cloud but APP does not is problematic and is the one we want to minimise. Over the 17 years common to both products, restricting ourselves to the period 1 July - 31 October studied in this manuscript, this case happens on average to 19.1% of the pixels (supp. Table A1). The opposite case, where a clear pixel is wrongly detected as cloudy by APP and eliminated from the study, happens on average to 12.4% of the pixels (supp. Table A1).

We investigated the possible cause for these misdetections by randomly plotting and scrutinising individual days. We here present only one of them, 10 September 2014, as it is relatively easy to analyse even to the untrained eye (supp. Fig. A1). We

**Table A1.** For all years from 2002 to 2018, between 1 July and 31 October (dates of winter for polynya detection in this manuscript), median percentage of pixels: that are cloudy according to the reference cloud mask MYD35_L2 but were not detected by the criteria applied to APP (first column); that are clear in the reference, but were incorrectly detected as cloudy in APP (second column); which are detected in both products as clear or cloudy (third column). First line is when applying the original Yamanouchi et al. (1987) criterion $|T34| > 2$ K; second line, when modifying it as explained in this appendix.

|  | Cloudy in MYD35_L2 | Cloudy in APP | Both agree |
|---|---|---|---|
| $|T34| > 2$ K | 19.1% | 12.4% | 64.5% |
| $|T34| > 1.5$ K | 14.4% | 15.5% | 67.4% |

focused on the pixels where the reference is cloudy but APP is clear (black dots on supp. Fig. A1c and d). These pixels fall in two categories:

- they are on the edge of a cloud, and hence may have moved between the acquisition of MY35_L2 and that of APP;

- or they correspond to $|T34| > 1.5$ K (highlighted with yellow core on supp. Fig. A1d).

The first case cannot be further verified, as APP is composed of a mosaic of images acquired over a 6h window. For the second case however, we modified the criterion for cloud detection in APP from $|T34| > 2$ K (Yamanouchi et al., 1987) to $|T34| > 1.5$ K, and performed the comparison again. The improvement was significant, with "missed" pixels decreasing from 19.1% to 14.4% overall (supp. Table A1). The agreement between the two products even increased from 64.5% to 67.4%. By decreasing the T34 threshold we increased the sensitivity of APP, resulting unfortunately in an increase in the number of pixels wrongly detected as cloudy from 12.4% to 15.5%.

In conclusion, we apply throughout this paper the modified criteria for clear pixels:

- T4 $\geqslant$ 245 K (Yamanouchi et al., 1987);

- $|T34| \leqslant 1.5$ K (after Yamanouchi et al., 1987, and this appendix);

- and 0 K $\leqslant$ T45 $\leqslant$ 2 K (Vincent, 2018; Vincent et al., 2008).

For the entire period common to the two products, these criteria fail to detect 14.4% of the pixels as cloudy, but also wrongly detect 15.5% of the pixels as cloudy when they are clear. As these two values balance out and that they are, to some extent at least, caused by the difference in acquisition time of the two products, no further action is taken to estimate the accuracy of our results.

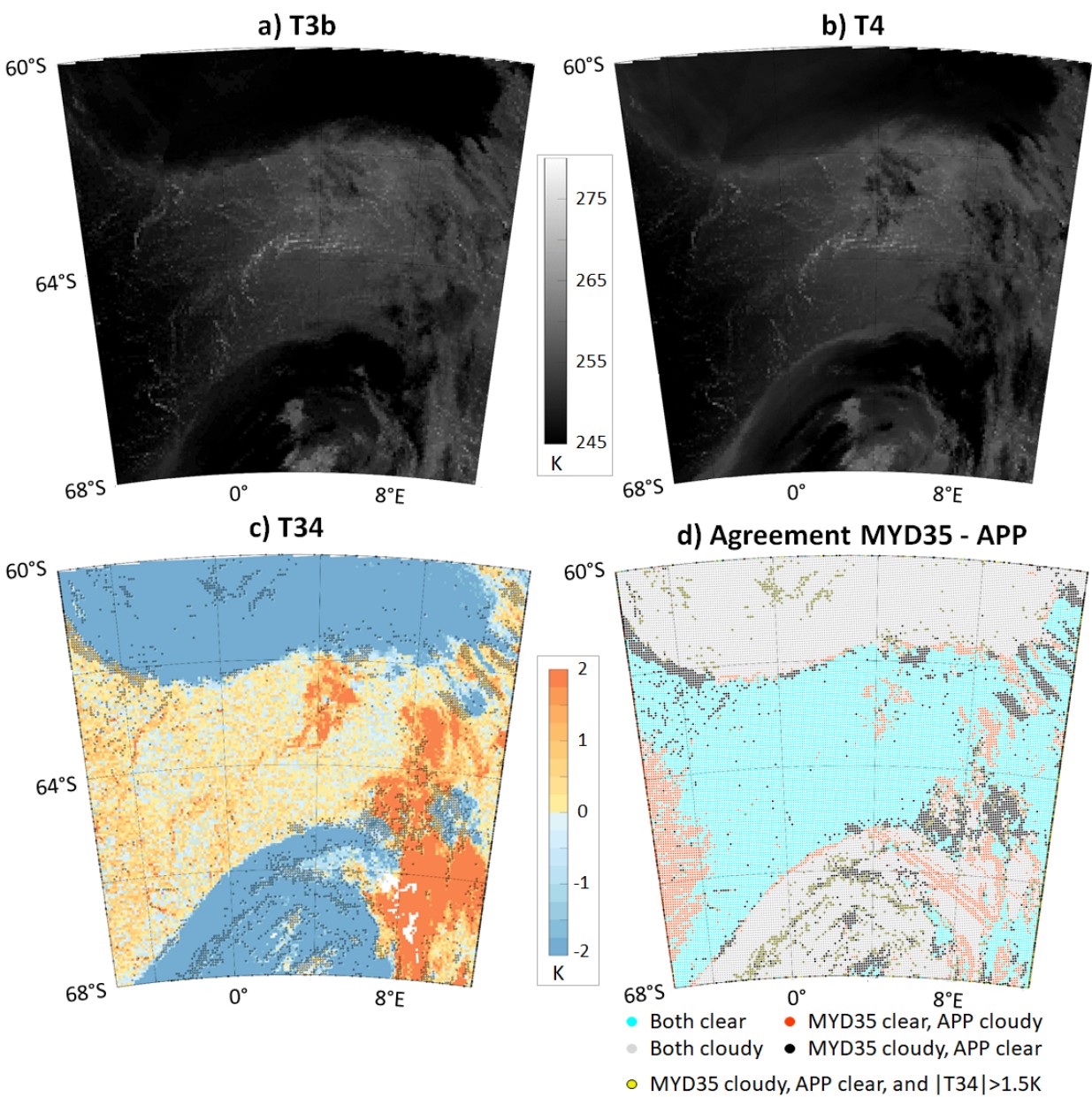

**Figure A1.** 10 September 2014. Infrared brightness temperatures T3b (a) and T4 (b), on the same colour scale, and their difference T34 (c). On these three panels, two large cloud systems are visible at the top and bottom (low T34), and smaller clouds to the right (large T34). Comparison of the cloud mask based on the APP data and the MYD35_L2 reference (d): cyan for light pixels in both, grey for cloudy pixels in both, orange for pixels clear in the MYD35_L2 reference but detected as cloudy with APP, and black for pixels cloudy in the reference but incorrectly detected as clear. These black pixels are also indicated on (c). A yellow core highlights these incorrect black pixels where the absolute value of T34 is larger than 1.5 K, i.e. that are correctly detected as cloudy when the threshold on T34 is lowered from 2 K to 1.5 K.

**Appendix B:  Supplementary tables and figures**

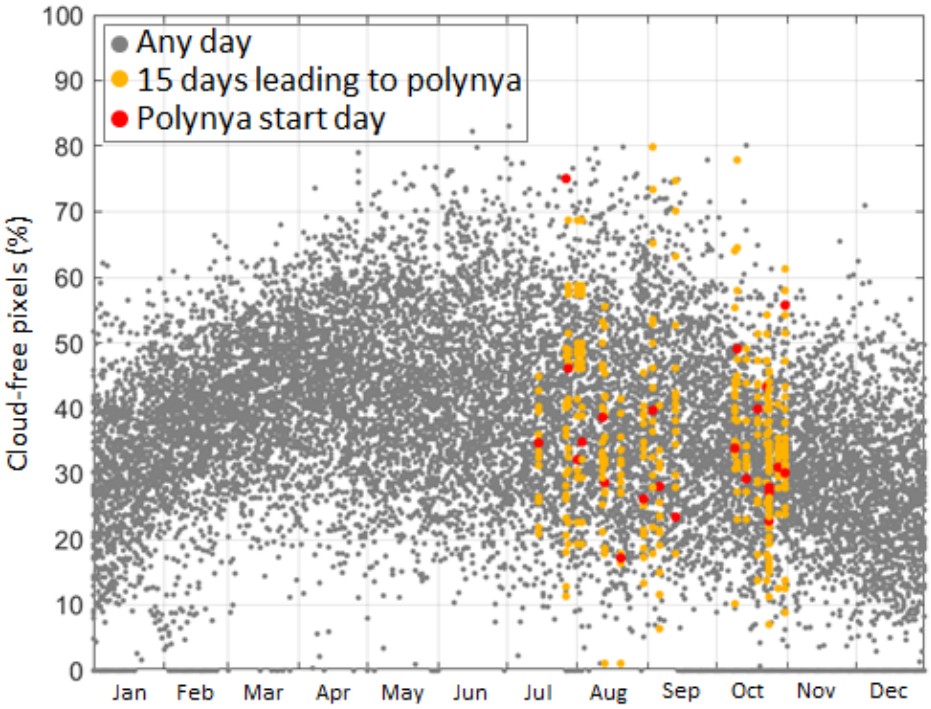

**Figure B1.** For each day of the year, percentage of cloud free pixels (out of 90000) for any year sampled by APP (grey), for the 15 days leading to a polynya event (orange), and the days a polynya opens (red).

**Table B1.** Characteristics of the events detected in section 3.1, with sea ice concentration < 60% criterion: start date, latitude (lat., in degree North) and longitude (lon., in degree East) of the centre, maximum area (in $km^2$) and duration (in days). The last two columns are the median percentage of cloud-free pixels (out of 90000) over the 15 days prior to the polynya opening, as well as the minimum-maximum interval of that percentage.

| Start date | centre lat. | centre lon. | max. area | duration | cloud-free, median | cloud-free, min-max |
|---|---|---|---|---|---|---|
| 17 July 1980 | -63.9 | 2.2 | 5625 | 3 | N/A | N/A |
| 15 July 1991 | -64.1 | 4.7 | 12500 | 5 | 34% | 21-45% |
| 24 Oct 1991 | -63.7 | 5.1 | 5000 | 8 | 27% | 16-41% |
| 23 Oct 1992 | -63.8 | 5.4 | 25625 | 9 | 23% | 8-35% |
| 13 Aug 1994 | -63.7 | 3.2 | 21875 | 6 | 33% | 2-56% |
| 20 Aug 1994 | -63.9 | 4.2 | 7500 | 5 | 30% | 2-42% |
| 6 Sept 1994 | -64.0 | 5.0 | 18125 | 15 | 24% | 7-41% |
| 18 Oct 1996 | -63.4 | 5.6 | 9375 | 11 | 38% | 14-52% |
| 22 Oct 1996 | -63.7 | 3.7 | 9375 | 7 | 41% | 14-55% |
|  | -63.6 | 6.6 | 625 | 1 |  |  |
| 30 Oct 1996 | -63.4 | 5.6 | 625 | 1 | 43% | 14-62% |
| 27 July 1999 | -67.7 | -4.4 | 9375 | 1 | 30% | 12-76% |
|  | -66.6 | -0.2 | 10000 | 2 |  |  |
| 24 Oct 1999 | -66.4 | -0.2 | 14375 | 8 | 38% | 18-46% |
| 29 Aug 2000 | -66.4 | -1.3 | 21875 | 3 | 34% | 14-46% |
| 27 Oct 2000 | -65.1 | 1.0 | 1250 | 2 | 32% | 13-41% |
| 30 Oct 2000 | -65.2 | 1.8 | 6875 | 2 | 31% | 9-36% |
|  | -64.2 | 6.8 | 14375 | 2 |  |  |
| 9 Oct 2004 | -67.6 | 4.9 | 5000 | 4 | 39% | 24-78% |
|  | -63.8 | 5.7 | 8125 | 17 |  |  |
| 13 Oct 2004 | -67.5 | 2.0 | 4375 | 2 | 33% | 24-50% |
|  | -66.6 | 5.6 | 7500 | 2 |  |  |
|  | -63.7 | 1.7 | 33125 | 14 |  |  |
| 9 Oct 2005 | -63.9 | 2.2 | 1875 | 2 | 40% | 11-65% |
| 27 July 2016 | -64.2 | 6.5 | 2500 | 3 | 47% | 20-69% |
| 31 July 2016 | -64.1 | 5.7 | 625 | 1 | 47% | 20-69% |
| 2 Aug 2016 | -64.3 | 6.3 | 43125 | 8 | 47% | 20-69% |
| 11 Aug 2016 | -65.0 | 3.9 | 17500 | 4 | 35% | 22-54% |
| 3 Sept 2017 | -63.9 | 3.7 | 3125 | 5 | 42% | 18-80% |
| 13 Sept 2017 | -64.3 | 4.2 | 71875 | 49 | 41% | 18-75% |

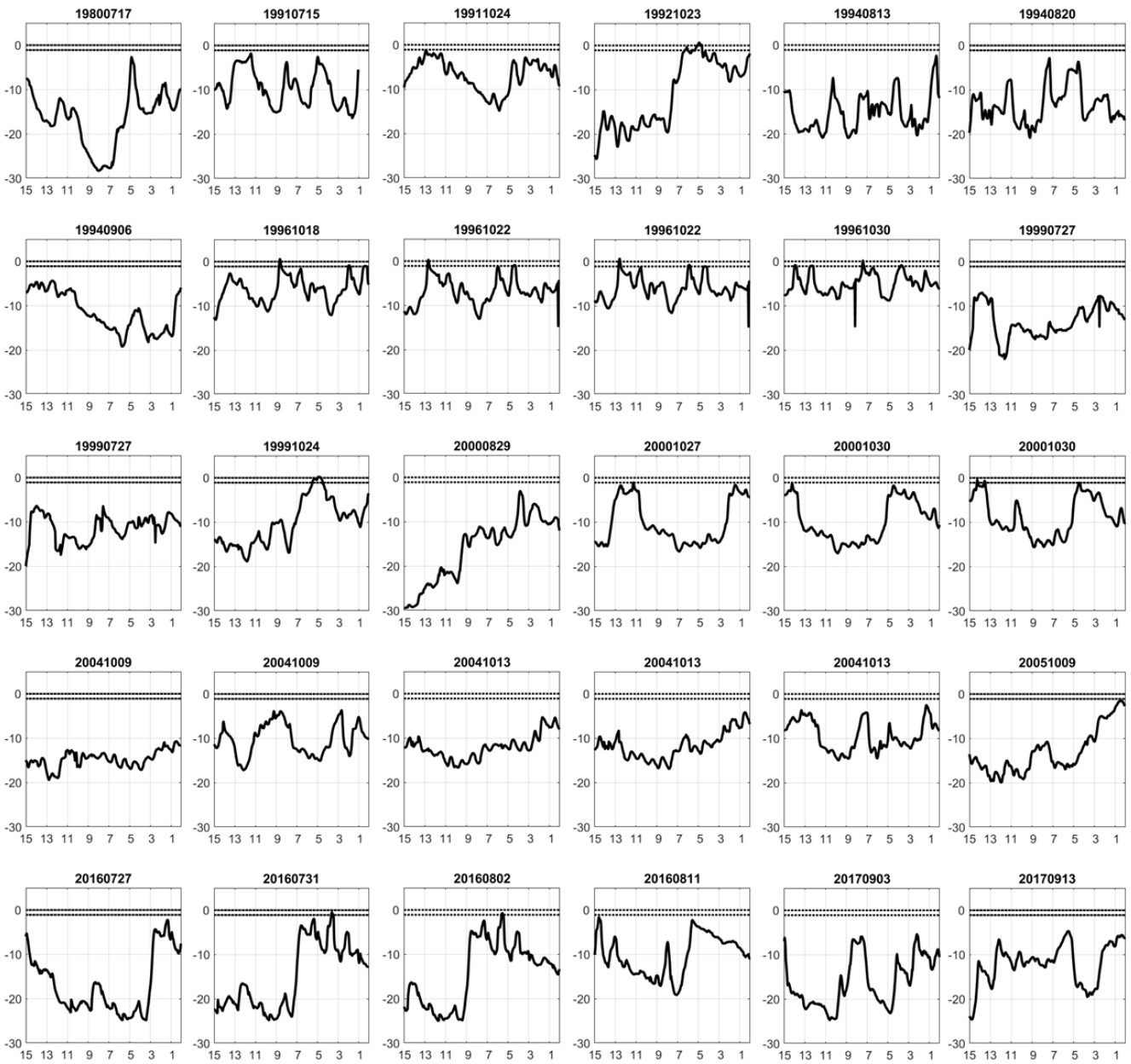

**Figure B2.** Time series of the 2 m air temperature at the location of the polynya, over the 15 days leading to the polynya event (date of the event in the panel title). x-axis: Days before the polynya event; y-axis: Air temperature in degree C. Horizontal dashed lines indicate the freezing temperature for young salty (-1.1 °C) or older fresh (0 °C) sea ice

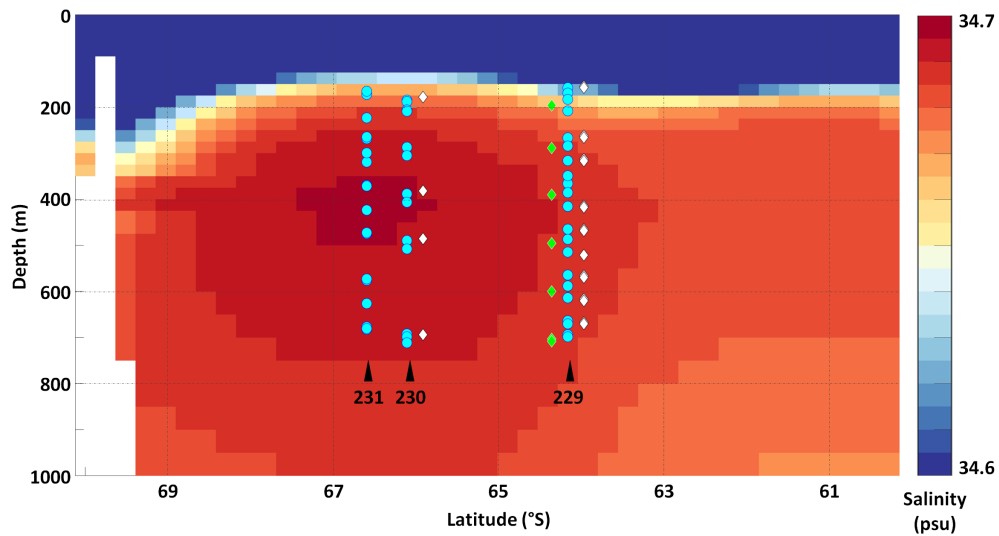

**Figure B3.** Climatological salinity along the Prime Meridian, from Zweng et al. (2018), showing the comparatively salty CDW in red. For each mooring line (229, 230, and 231), the symbols indicate the depth of the mooring hydrographic sensors during deployments that coincided with polynya events: white diamonds for 1999 and 2000; cyan circles for 2004 and 2005; and green diamonds for 2016. The latitude shift between deployments of the same mooring is for readability only.

**Table B2.** Left: median pressure (dbar) of the shallowest temperature sensor on each mooring line (columns) for each year over the period 1st July - 31st October. Middle: correlation coefficient over that same period between the temperature (T) of that shallowest sensor and the infrared brightness temperature T4 at the same location. Right: correlation between the salinity (S) of the shallowest sensor and T4. Only significant correlations are indicated; insignificant correlation at the 95% confidence level are indicated with a '-'; absence of sensor or data, with 'N/A'. Years with a polynya event are highlighted with an asterisk *.

| Year | Pressure 229 | Pressure 230 | Pressure 231 | Correlation T 229 | Correlation T 230 | Correlation T 231 | Correlation S 229 | Correlation S 230 | Correlation S 231 |
|---|---|---|---|---|---|---|---|---|---|
| 1996 | 203 | 80 | 206 | 0.28 | 0.26 | 0.31 | 0.25 | N/A | N/A |
| 1997 | 204 | 87 | 206 | 0.08 | 0.11 | - | 0.11 | N/A | N/A |
| 1998 | 187 | N/A | N/A | 0.09 | N/A | N/A | 0.22 | N/A | N/A |
| 1999* | 154 | 178 | N/A | 0.22 | 0.37 | N/A | 0.22 | N/A | N/A |
| 2000* | 158 | 178 | N/A | 0.25 | 0.30 | N/A | 0.24 | N/A | N/A |
| 2001 | 225 | 196 | 253 | 0.36 | - | 0.26 | 0.32 | N/A | 0.57 |
| 2002 | 235 | 196 | 253 | 0.19 | - | - | 0.19 | N/A | 0.15 |
| 2003 | 159 | 185 | 171 | 0.38 | - | - | N/A | N/A | N/A |
| 2004* | 158 | 185 | 171 | 0.09 | 0.11 | 0.13 | N/A | N/A | N/A |
| 2005* | 183 | 206 | 166 | - | 0.29 | 0.11 | - | N/A | 0.18 |
| 2006 | 187 | 206 | 159 | 0.22 | - | - | 0.20 | N/A | - |
| 2007 | 324 | 206 | 158 | 0.22 | 0.12 | - | 0.29 | N/A | - |
| 2008 | 191 | 262 | 221 | 0.27 | 0.11 | - | 0.28 | N/A | 0.17 |
| 2009 | 192 | 262 | 221 | - | - | 0.23 | - | N/A | 0.32 |
| 2010 | 195 | 262 | 221 | 0.14 | 0.16 | - | 0.17 | N/A | 0.39 |
| 2011 | 692 | 218 | N/A | 0.16 | 0.09 | N/A | N/A | N/A | N/A |
| 2012 | 694 | 426 | N/A | 0.11 | - | N/A | N/A | 0.19 | N/A |
| 2013 | 213 | 221 | 228 | 0.24 | 0.32 | - | N/A | N/A | N/A |
| 2014 | 216 | 221 | 228 | 0.12 | 0.48 | - | N/A | N/A | N/A |
| 2015 | 197 | N/A | N/A | - | N/A | N/A | - | N/A | N/A |
| 2016* | 197 | N/A | N/A | 0.26 | N/A | N/A | 0.28 | N/A | N/A |
| 2017* | 314 | N/A | 4629 | 0.09 | N/A | 0.14 | N/A | N/A | - |
| 2018 | 353 | N/A | 4629 | 0.55 | N/A | 0.09 | N/A | N/A | 0.13 |

*Author contributions.* C.H. designed the study and conducted the Infrared and Mooring analysis with L.Z; M.M. conducted the float analysis; A.L., the SAR analysis which was eventually not included. All authors contributed to the manuscript.

*Competing interests.* The authors declare no competing interest.

*Acknowledgements.* This project is funded by the Swedish National Space Agency (grant 164/18 awarded to C.H.). Data were collected and made freely available by the Southern Ocean Carbon and Climate Observations and Modeling (SOCCOM) Project funded by the National Science Foundation, Division of Polar Programs (NSF PLR -1425989), supplemented by NASA, and by the International Argo Program and the NOAA programs that contribute to it. The Argo Program is part of the Global Ocean Observing System (http://doi.org/10.17882/42182,
http://argo.jcompos.org). The authors thank Thomas Lavergne for his generous help while this idea was still at the proposal stage, and again, along with his colleagues at Metno, for their help with getting the data and eventually suggesting we use APP. We thank Julia Kukulies and Hui-Wen Lai for their help with the bit stripping script for the MYD35_L2 cloud masks, Anna Wåhlin for explaining so clearly mooring line movements, and Ethan Campbell and Dana Swift for their detailed explanation of the ice-avoidance algorithm. We also thank the two anonymous reviewers 1 and 3 and Stefan Kern (reviewer 2) along with the editor Jennifer Hutchings for their comments that greatly improved
the clarity and quality of this manuscript. Finally, C.H. is grateful to Le Chat for his unconditional support during these long working-from-home months.

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
