# Peer review of "Spaceborne infrared imagery for early detection of Weddell Polynya opening"

_The Cryosphere, 2020_

## Referee Comment (RC1) · Anonymous Referee #1 · 27 Aug 2020

Summary:

This paper develops an early opening detection method for the Weddell Polynya using the spaceborne infrared imagery, AVHRR. Causes of the openings are investigated using atmospheric reanalysis, hydrographic mooring and Sentinel-1 SAR data. The results indicate that the method performs well on detecting the events without finding false positives. Both oceanic upwelling and wind-induced divergence contributes to the opening of Weddell Polynya.

General comments:

1. The summary of satellite observations for sea ice monitoring needs improvement. A

more thorough reading of the relevant references is needed.

2. The method is not clearly described in the manuscript. A flowchart of the "early warning system" could be helpful. For example, how the polynya-prone area is determined and how it varies with years? In Section 3.1, when determining the polynya dates, is the minimum SIC area-wise (the polynya-prone area)? If yes, is the polynya-prone area fixed? How large? How many pixels are we talking about? Such questions pop when we start to read section 3.1, however only find it till the end (in the caption of Figure 3). In section 3.2, why plotting the data on the 15-day minimum and maximum axes? It is not clear that, what are the final criteria for the early detection until I read the manuscript a couple times.

3. It is a bit difficult to follow the flow of the manuscript, especially for the method. Section 3.1 and 3.2 are part of the method however presented in the "Results and Discussion" section. It could be easier to read if restructuring the "Method" and "Results and Discussion" sections, with "the warning system" (and identifying criteria) in "Method" and "cause of the opening" in "Results and Discussion".

Specific comments:

P1L1: "a crucial information" to "crucial"

P2L45-46: "sea ice concentration from passive radiometers or spectrometers (Spreen et al., 2008), mostly in the microwave region ..." I don't recall any spectrometers used for sea ice concentration retrieval. The most commonly used data to derive sea ice concentration is microwave radiometer data, because the sensor (i.e. microwave radiometer) can work independence of light and penetrate clouds. There is no need to mention spectrometers. It can be rewritten to sth like "... sea ice concentration from microwave radiometers (Spreen et al., 2008)."

P2L46-52: The authors give a list of references to present the low and high resolution applications. It is true that SAR has much higher spatial resolution than microwave

radiometer, BUT both are microwave sensors. It is not appropriate to use "microwave products" to refer to the radiometer/scatterometer-based products. On the other hand, it is common to use microwave radiometers for sea ice drift tracking, sea ice classification and thickness retrieval (e.g., SMOS/SMAP). In Korosov et al., 2018, the microwave radiometers were firstly used to derive sea ice drift information, on the basis of which sea ice age was tracked. These sentences should be rephrased. Please also check the use of "microwave" (when referring to microwave radiometer), in the whole manuscript.

P2L52-53: Microwave radiometers have been used to monitor sea ice since late 1970s, they are not recent sensors compared to infrared sensors.

P3L59: "determine the mechanisms" to "understand the mechanisms"

P3L69: "reanalysis"

P3L72-74: since these two datasets have different resolution, please specify the resolution respectively.

P6L128-129: "using the traditional criterion sea ice minimum . . .", do you mean sea ice concentration minimum? It is not clear here how the "sea ice minimum" is determined? If this is the minimum concentration of the polynya-prone area, a description of how the polynya area is determined and how large is the area is needed.

P7L140: "yield" to "yields"

P7L143: "then a third opens . . . the largest one splits into two". It is confusing here, which one is the largest one?

P7L147-148: "thirty" to "30"; ""

P7L166: 3443 days? For how many years? In the next paragraph, the number of days is replaced with the number of events? Why?

P7L160-171: These two paragraphs basically explains that threshold for single band does not work well for early detection but the three bands together do. A summary of

the final criteria and the reason is needed. P12L220: which three events do you mean? Do you mean the average temperature in the 15 days, or when the opening starts? In Figure A5, there are more cases with 2 m air temperature of -10 degree by the time of opening.

P13L235: I don't find temperature and salinity at 480 m depth oscillate with a near 12 hour frequency, at least not from -30 days to -13 days. Only smaller oscillations are found during 30-20 days before the event.

P13L250: Again, I don't find increase of temperature at depth. It is too small to notice. The brightness temperature increases following the dip of T45 however only for one day.

P7L257-258: I would suggest delete the two question mark and words in parentheses.

P15L265: "rejected"

P17L300: "Sentinel" to "Sentinel-1", the revisit time for the twin Sentinel-1 satellites is 6 days not 4 days.
* * *

---

## Referee Comment (RC2) · Anonymous Referee #2 · 23 Sep 2020

Review of

Spaceborne infrared imagery for early detection and cause of Weddell Polynya openings

by

Heuze, C., and A. Lemos

abstractSummary The authors attempt to close a discussion about the trigger of the open ocean Weddell Polynya, whether it is predominantly a latent or a sensible heat polynya. Their result at the end is that it is both, suggesting to close the discussion. The authors

reach to that finding by first identifying potential polynya events by means of passive microwave sea-ice concentration data and subsequently merging that information with some statistics derived from gridded infrared brightness temperature (IR TB) observations of AVHRR channels 3, 4 and 5. By combining this information with ERA5 atmospheric reanalysis data of the 2m air temperature and the 10-m surface wind, deriving the CURL, the authors attempt to find criteria to be able to predict a polynya event. In addition to this analysis, the authors combine their results for a few cases with hydrographic observations by moorings along the prime meridian. Based on a qualitative discussion of the used observables the authors end up at their conclusion.

While this manuscript includes some interesting measures and could indeed shed more light on the triggering mechanism of the open ocean Weddell Polynya, their general concept to define when such a polynya developed is not conclusive and bears the danger to mix these true open ocean polynya cases with cases of a comparably high concentration of leads.

I have several methodological concerns which possible are borne out of the fact that the description of several aspects of the data processing is overly light - as referred to in my general comments GC1 and GC2. Neither do I find the application of the AVHRR data to define preconditioning for polynya events convincing nor is, to my opinion, the discussion about the influencing factors complete. A few key investigations appear to be missing.

While I find the merging of satellite with the mooring data to the point in terms of that one should do it, the discussion could not convince me. Too many ambiguous signals are presented by the time series shown. While one could argue that exactly this ambiguity is finally translating into the main result of the paper, the fact that the authors could not convince with methodology, interpretation and discussion of caveats and limitations made me to seriously doubt that the way the paper is written is providing the hypothesized result.

General comments (GC)

GC1: I have a general concern with the APP-X data used and in particular with their quality. Since this is a key data set of this paper it deserves a much more detailed consideration, especially with respect to clouds which masking is, to my opinion, not convincingly illustrated. Here the manuscript requires some additional figures / illustrations as well as some re-writing.

GC2: I also have a general concern with section 2.3 which I find too generic. A scientist wishing to repeat you study would be lost because the description of the methodology is not adequately detailed. Here the manuscript requires some re-writing. In addition, I have the impression that choosing a different SIC algorithm would result in different "polynya events"

C3: I have a general concern about the applicability of the various infrared TB parameters. The caveats of the parameters used to delineate polynya conditions or conditions leading to the formation of a polynya are not sufficiently well laid out. Taking the statistics of IR TBs over an about 500 000 sqkm large region as a measure to better delineate the presence (or to-bt-presence) of a polynya of a size 1% of that area appears not to be well motivated.

C4: I have a general concern about the interpretation of the various infrared TB parameters - particularly in the context of figures 7 through 9 when it comes to the examples. The interpretation of observed changes in these IR TB parameters appears to be biased towards oceanic and sea-ice conditions, leaving out the direct influence of the atmosphere on the sea ice / snow surface. By the same token, a critical discussion of the usefulness of the wind data for the curl computations is missing.

Specific comments

Lines 17-21: I doubt that motivating the importance of the Weddell Sea polynya, which is a winter phenomenon, with commercial exploitation of ice-infested waters is a correct

thing to do for the Antarctic. Most tourism is in summer; fishing occurs outside the sea-ice cover; transportation does not happen, and exploration is limited thanks to the Antarctic Treaty. Hence, it is rather an interesting bio-geo-physical event with highly interesting ocean-ice-atmosphere interactions than anything else.

Lines 44-57: This paragraph requires some rewriting; the following issues are either not correct / not approriately formulated and/or referenced

- Line 47: Satellite passive microwave data are used to derive sea-ice motion. Based on the sea-ice motion sea-ice age is derived; while Korosov et al. (2018) worked on a long-needed adjustment of the motion-to-age conversion the original work dates back to 2003 or 2007, I guess. Maslanik or Tschdui should be the main authors, citations you can find in the paper by Korosov et al. (2018).

- Line 48: The algorithm proposed by Spreen et al. (2008) was based on AMSR-E data, the predecessor of AMSR2. That satellite did not allow for the stated fine resolution of about 3 km. One of the few citations pointing towards that there is a version of the same algorithm applied to AMSR2 data which allows for about 3 km grid resolution would be the one of Beitsch et al. (2014) in the journal "Remote Sensing", vol. 6.

- Lines 49-51: If you keep them, then all these citations require an "e.g. in front of the author(s) because there is a full suite of papers into these directions ... except the one for melt ponds; that one I suggest to delete because if you read that carefully you will figure out that the way they tried to infer melt ponds from HH-Pol wideswath SAR imagery did not work out well. While there have beeen many studies involving SAR and melt ponds I doubt THE solution has been found here and therefore it is for sure NOT the norm to use SAR for melt pond detection. Furthermore SAR and sea-ice thickness retrievals is also something I would not put under the "norm" to use SAR imagery. The range of maximum sea-ice thickness values to be retrieved using SAR goes from 0.1 m to 2 meters; this is a field which has not yet been explored enough and we are also not yet there - despite what machine learning and other similar approaches attempt to sell

us. SAR imagery is extremely ambiguous and not really suited for sea-ice thickness retrieval.

- Line 52: "microwave and SAR" –> SAR is microwave. "comparatively recent sensors" mentioned together with a 20+ years old citation does neither look overly smart nor is it correct, because microwave sensors date back to 1972/1973 with ESMR being the first one to allow to observe sea ice independent of clouds and daylight - being the sensor to allow detection of the Weddell Polynya for the first time.

- Line 55: "hence has fallen ..." –> Same comment as above in terms of 20+ years old citation plus: There has been quite some work with spaceborne infrared sensors during the past two decades. MODIS, for instance, has been used heavily for polynya detection in both hemispheres. It has been used for fast ice detection in the Antarctic. AVHRR and nowadays VIIRS has been used for sea-ice cover retrieval and, more importantly thin ice thickness retrieval quite extensively as you may note from papers by Key et al., 2016, The AVHRR polar pathfinder climate data records, Remote Sensing, 8 or Mäkynen and Karvonen, 2017, MODIS sea-ice thickness and open water-sea ice charts over the Barents and Kara Seas for development and validation of sea-ice products from microwave sensor data, Remote Sensing, 9. It might therefore make a lot of sense to better embed you work into the current state-of-the-art after you have undertaken a little literature review.

Introduction: I am sure a reader would appreciate a short paragraph in which you very briefly describe which data sets you applied to achieve which findings so that the list of data which follows falls on well-prepared ground.

Section 2.1:

- I suggest to provide a table in which you summarize which data you use at which spatio-temporal sampling for which time period. Also the grid information (i.e. which grid is used) should be included.

[Figure]

- Line 67 and line 73 contain contradicting time information: 1980 vs. 1978.

- There is a more recent version of the APP data set called APP-X. Did you use the latter one?

- I note that you provide a doi for the ERA5 data but not for the other data sets? Aren't there doi's for other data sets as well?

- I note that your description of the data is quite light and poses questions.

a) Which algorithm is the long-term sea-ice concentration data set from NSIDC based upon. If it is the NASA-Team algorithm then you need to be aware of the events of substantial underestimation of the sea-ice concentration due to snow metamorphism which this algorithm is subject to in late winter / spring. Other algorithms and products, i.e. the Comiso Bootstrap v3 or the OSI-450 data set do not have these problems; particularly the latter one, which is a CDR, appears to be most suited for your purpose.

b) Which resolution has the "product from the University of Bremen"? On which algorithm is it based? Over which time period did you use these data? Did you consider to use the 3.125 km product offered by the University of Hamburg (now from AWI) being available since 2012 based on AMSR2?

c) "Hydrographic data" is quite generic. Which data measured how at which depths over which time periods with which instruments?

d) Even though you used SAR only for qualitative purposes it would be important to learn how many you used, which time-differences these have with the other remote sensing data, and so forth. If it is just a dozen or so you can simply provide a table with overpass time, time difference to whatever other data, et cet. In addition "backscatter information" is also very generic. What is the purpose and what were you after in particular. One sentence stating that you were trying to find open water / new ice signatures in C-Band HH backscatter images would possibly be enough.

e) The AVHRR data set used ... is this an FCDR or CDR? Is it inter-sensor bias

corrected? I am asking because it is known that the various individual satellites hosting AVHRR sensors have been prone to serious drift in their orbits, shifting local overpass times by up to several hours.

f) Line 80 "For validation" –> What did you evaluate? Please add.

Section 2.2:

I suggest to considerably expand this section. The illustration of the cloud mask is not convincing. Since the detection of the polynya and/or its preconditioning conditions is among the key elements of this paper and since these infrared data are the key data set to be used this section deserves a lot more attention to ensure that any conclusions taken are not simply taken because of either an insufficient cloud mask or a drift in the cloud mask capabilities to mask out clouds or a drift in the used APP-X product.

One way to check the cloud mask would be to show coincident high-resolution MODIS and/or Landsat images. It could be sufficient to have the reflectances only because structures and textures evident in the reflectance and the infrared images might be similar. In any case would it be much better to actually see several cases where your approach correctly allows to delineate the surface temperature - i.e. by showing the temperature gradient across the ice edge or within a polynya area. Currently, this section is not convincing - neither about the usefulness of the APP-X IRBT data nor about the cloud masking scheme.

In this context, it might make sense to look at the recent paper of Vincent et al. from 2019 in Remote Sensing, 11, "The case for a single channel composite Arctic SST algorithm".

Section 2.3:

- Line 105: Please specify in your text when and why you use a per-pixel SIC threshold or an area-average threshold. It is not clear why both are used and/or required.

- Line 107: "we tested .." –> Very generic description. Please provide more details in

your text. How did you visually assess each option? Against which information did you do this assessment? Over which period and with which frequency of the input data did you carry out this testing?

- Line 108: "contiguous pixels" –> How did you assure / identify whether pixels are contiguous or not? Please provide more details in your text.

- "pixel area" –> What did you use as the pixel area. If you have used data on a polar-sterographic projection, which I assume you did, then you should have taken the grid-cell area of each individual grid cell from a separate data frile provided by NSIDC. This needs to be clarified.

- Line 109: "60% for most of the study" –> What are the incidences where you needed to use a different threshold and why? Please be more specific in your text.

- Please provide information about the time period per year you use. Currently one can think that you try to do the analysis for the entire year (which of course is not logical because of the lack of sea ice in the polynya-prone region for a number of months. But you don't tell the reader your time period of interst.

- Line 105-110: For what is this data set created? What is the purpose and how ist it connected to the other parts of the method?

- Line 113: "For robustness though ..." –> It is not clear from your text why this step is needed to check the robustness. It is also not clear what you mean by brightness temperature composites. This requires further clarification if kept.

- Lines 115/116: The "anomalies" you are writing about, did you compute this separately for all parameters, i.e. did you compute daily climatological values for median, minimum and maximum IR TB?

- Line 116: "over the 40 years" –> It is 37 years, isn't it?

- Line 117: "using only the years ..." –> Is this surprizing? Possibly not because if I

understood your introduction / motivation correctly it was 2016 for the first time that the Weddell Polynya opened again within the considered 1982-2018 period ... hence the majority of the data are not including polynya conditions anyways.

- Line 119: "over each polynya" –> I though you are only targeting the Weddell Polynya so which other polynyays referred to by "each" are you investigating here? It is not clear from your writing why there should be several polynyas.

- Line 125: Please provide a reference for this interplay between convergent situations and upwelling.

Figure 2 / Line 130: What motivates using a mean SIC threshold of 92%?

Line 137: Which "other methods" are you referring to here?

Line 139: "visual validation" –> I would certainly not term this "validation"; I'd rather say you did a plausibility check. How did you assess the 60% criterion quantitatively? Is there any quantitative information you can give young students and/or scientists at hand who want to repeat you study?

Table A1: Did you check you analysis with respect to the credibility of the polynya events detected? Looking at Table A1 I am concerned with cases where the polynya (according to the 60% criterion) lasted just 1 or 2 days but occupied a comparably vast area of nearly 10000 sqkm. How realistic do you consider such a rapid opening and closing of the polynya?

Line 151: Following up with my earlier comment about which sea-ice concentration product you used I recommend to repeat the same analysis (and in particular the numbers shown in Table A1) using a different algorithm, e.g. OSI-450 or Comiso Bootstrap v3.1. This way you would be able to considerably enhance the credibility of your findings and eventually eliminate the cases of a overly rapid opening / closing of polynyas of considerable size.

Lines 156/157: "As explained ..." –> where exactly did you made statements in the

methods section backing this up?

Line 162 / Figure 4:

- "whereas it is very specific for T4" –> I doubt that this statement should get any weight. The fact that channel 4 minimum IR TB is constant at 245 K for all cases investigated points to at least an issue with the cloud mask if not to a completely non-credible TB value. What would be the physical explanation for the observation that the minimum channel 4 IR TB for the polynya-prone region (red rectangle in Fig. 3) is the same across all 14-day periods prior to the dates shown in Table A1?

- I note that the maximum channel 4 IR TB of the 14-days prior to the days given in Table A1 exceeds 0degC. What is the physical explanation for this? Having +5degC for an IR temperature translates into a physical temperature close to +10degC. Is this realistic? I doubt so.

- How would Fig 4 look like in case you would have used the same (by yearday) 14-day periods for the same area (or grid cells) but from years without a polynya event? That way you would have real inter-comparison measure at hand. I guess, by using the anomalies you attempt to go into this direction. However the data clouds in the anomalies for median and minimum appear to be trivial, pointing to linear relationships with an approximate equal share between positive and negative anomalies for the median and of course larger negative deviation for the 15-day minimim than the 15-day maximum for the geographical minimum. Thanks to the quite dubious distribution of the daily IR TB minimum values in the left panel and the several > 273 K daily IR TB maximum values I would not trust the anomalies for the geographical minimum and maximum the way presented anyways ... . Hence, Figure 4 and its description / interpretation should be re-written and clarified.

Lines 165-168:

- "by using their smallest value as a threshold" –> to do what? This is not clear. Please

be more specific in your description.

- "There are 3443 days ... " I don't understand what you did here? Please re-phrase.

Line 169: Why "sacrifice"? I don't understand. Please re-phrase.

Line 172: I still don't understand how you reach to the "36 false positives" because it is not clear what you did with which IR TB (threshold) value applied to which data set.

Line 174: What is the difference between the "size of the polynya" and the "footprint on the IR TB data"? I don't get it. This needs to be explained. I note in this context, that an illustration of some examples (in general), i.e. how the SIC map and the AVHRR channel 3,4,5 IR TB maps look like in case of a small / large / poorly or well-defined polynya, would greatly improve readability of the paper and credibility of the method and the results. The method is not well illustrated.

Table 1: I am a bit confused about usage of the University of Bremen data set (just?) here in the context of better explaining the false positives. This invokes a certain degress of inconsistency in your work.

Line 185: I don't agree to this statement. By what you showed in your paper so far you did not convince that the method works robustly. Still I will take it as granted and continue on with my review.

Lines 189/190: It is not unlikely that there have already been leads in the pp-region before the Polynya events, isn't it? Regarding the T45 threshold I again refer to the more recent publication by Vincent et al. I mentioned earlier in my review.

Line 205: "the oscillation in (IR) TB suggests an upwelling of warm water" –> If I understood you correctly, then the examples of the oscillating IR TBs shown in Fig. 5 and in the respective figures in the Appendix are based on observations of the polynya-prone region (PPR) shown in Figure 3, extending south-north over 8 degrees latitude which is 480 nautical miles or about 800 km. Hence the oscillations shown reflect the conditions of a synoptic-scale region. It could (...) be that the oscillations and variations in IR TB

are caused by ocean upwelling. But wouldn't this provide that a substantial amount of the sea ice is i) bare = not snow-covered and ii) sufficiently thin so that a 1-day long upwelling event manifests itself in a change in the median or maximum IR TB of the PPR by up to 7 Kelvin? What is the typical temperature variation immediately beneath the ice bottom / at the water surface in the leads due to an upwelling event? I'd expect it is an order of magnitude smaller. What - to my opinion - is missing in your initial scientific reasoning here is the immediate impact of the atmosphere. Advection of warm or cold air can very well cause surface temperature changes of the observed magnitude within the PPR. The IR TB measured is direct measure of the surface temperature. To my opinion, the size of the PPR in combination with the magnitude of the observed IR TB changes and the fact that maximum IR TBs are often de-coupled from the median IR TB (which is fine as mild air might prevail in the north and cold air in the south of the PPR), confirms that a direct atmospheric influence is much more likely as a cause for the oscillations than oceanic upwelling. This is also supported by the knowledge that surface temperature changes caused by ocean upwelling require substantially more time than surface temperature changes caused directly by the atmosphere.

An issue being connected to this is that fact that we don't know how accurate the IR TBs are and how often cloud artifacts disturb the observations. IR TB artifacts due to clouds can create both comparably warm or cold signatures, depending on whether a low-altitude warm cloud or a high-altitude cirrus cloud was not adequately masked out. Furthermore, Figure 5 and the resepctive figures in the Appendix lack information about how many of the pixels of the PPR were actually cloud-free during the 14-day periods considered. My assumption would be that the number of cloud-free grid cells is considerably larger for the comparably cold median and maximum IR TB cases shown. In other words, the statistics of every data point shown in these figures potentially varies from day to day.

Figure A4 caption: I suggest to add the information that red (=positive) likely denotes divergent - aka - ice cover opens mechanically while blue (=negative) likely denotes

upwelling - aka - ice melt from below, or if there is sufficiently open water present already also lateral melt.

Line 212: "it can open the ice within 12 hours" –> Lets take this as a useful number for the moment. In this case, within one day, divergent wind conditions could result in a drastic increase in the observed IR TB because an area covered completely by sea ice of a surface temperature of -20degC might change in an area covered by 95% sea ice and 5% open water of -2degC, which would result in an area-average surface temperature of -19.1degC, an increase by about 1K. It is straight-forward to estimate other surface temperature changes as a function of open water created. Note that this simple consideration applies to a case where the open water does not freeze over again immediately.

Lines 214-221 / Fig. A5. This paragraph and interpretation of the results requires some re-writing.

- If I follow the considerations in the various Nandan et al. papers then I am inclined to say that surface melt of the snow cover would occur at 0degC most of the time - unless we talk about thin ice with a thin snow cover - which is something you don't know. Any thicker (above 6-8 cm) snow would possibly have zero surface salinity.

- I agree that the sea ice would not start melting at 0degC but already at a lower air (surface) temperature. However, the majority of the Antarctic sea-ice cover is snow covered.

- I am not sure what the intention of this paragraph is. Is it trying to use ONLY the air-temperature information to find a signal which could be interpreted as a pre-conditioning for polynya formation? This should be made clear in its first sentence.

- "The main caveat ... is colder" –> This reads as if the 2m air temperature is always colder than the surface temperature. I'd say the opposite is the case for snow covered sea ice - unless the surface was warmed by the mild air of a warm sector of a cyclone

and now a cold front brings cold air advection. Hence your statement as given might only be valid for young and thin bare ice or for the special case just described and needs to be re-phrased.

- "are reanalysis data" –> Sure. But you can specify this problem much better by figuring out how ERA5 treats sea ice. Guiding questions could be: Does ERA5 acculumate snow on sea ice? How is the effect of a snow cover parameterized in case there is none? How thick is the sea ice in ERA5? Also: You chose ERA5 on purpose. You could have used an alternative ice surface temperature product, e.g. from MODIS or the data set created by J. Comiso (et al.) from AVHRR data to get more confidence on your own IR TB and their relation to ERA5 data. Finally, you could cite inter-comparison studies involving ERA5.

I note that while you argue that ERA5 data might not be useful with respect to the air temperature you do not critically comment whether the curl derived from these data couldn't be unreliable as well and, e.g. provide the wrong magnitude or, more importantly, the wrong sign.

Figure 6: It there the possibility to show a similar plot for the salinity or at least an illustration of the typical vertical salinity profile during winter? This would aid in a better understanding that oceanic convection events would cause a decrease in T AND S at depth.

Line 229: "luckily, there were Sentinel-1 images" –> This information is out of context here because a SAR image cannot replace surface temperature or salinity measurements. I suggest to delete this information here and put it somewhere else at a more appropriate place.

Figure 7 and its discussion in lines 232-242:

- Given the concerns I voiced earlier with regard to the usefulness of the median, maximum or minimum IR TB parameters shown in panels a) and b), I would be reluctant

to interpret too much into the temporal relation between the observations at a point (the mooring) and a synoptic-scale region. What does an overall dip in the median IR TB in the PPR tell us specifically for the region where the polynya could form? In my eyes nothing more than that for the majority of the clear sky grid cells in the PPR (of which we neither know the number nor where these are actually located because of the cloud cover) the IR TB and hence the surface temperatures were low. Furthermore, any changes in the IR TB at PPR scale could (...) be due to changes in ice coverage and/or thickness caused by divergence and/or upwelling of warm waters but these changes have a large, if not dominating direct influence by the atmospheric temperature - which is not considered here.

- Are the hydrographic observations quality controlled? I am wondering about spikes in, particularly, the salinity which appear to be not realistic (see panel f), positive peaks on days -15 and -23, negative peaks on days -19 and -21 or panel e) day -16)

- The water temperature at 0 m depth is positive? How does this match with the climatology shown in Fig. 6. If this was the case, then the ice was constantly melting from the bottom.

- I recommend to use the same range for temperature and salinity in the three panels showing these observations to allow a better discrimination between noise and the actual signal. For salinity an appropriate range appears to be 0.05 psu. For temperature I suggest 0.25 K. That way it becomes clear that panel d) shows noise only.

- There is no signal of a water temperature increase and/or salinity change during the upwelling cases.

- Panel e) is closer towards the climatological warm core of the CDW than panel f) (compare Fig. 6); hence the temperature is overall larger in panel e) than f). This is fine. Assuming that upwelling occurs from this core, any upwelling events as indicated by the curl (panel c) should not be visible in panel f), am I correct? Interestingly, the upwelling event around day -20 is associated with no T or S change at depth 380 m

but with a T decrease (relative to the T observed during the strong / weak divergent conditions on day -21 / day -19) and a S decrease. Why? Is this upwelling from below the CDW core which only reached to that level but not to 380 m?

- Shouldn't changes in T or S associated with ice formation and water mass modification be observed first at 380 m and then at the deeper place of 480 m? The only event T and S variations at depths 380 m and 480 m are in phase are at day -16 when at both depths T and S had a relative minimum - pointing at the same time at an event which causes a decrease in T with a concomitant decrease in S.

- The upwelling event at day -12 has no signature in the hydrographic data.

- The partly pronounced divergent conditions from days -2 to -11 don't have a signature in the hydrographic data.

- The warming and concomitant salinification at days -6, -7 at depth 380 m is an isolated event, not supported by panel d) or panel f), and also not supported by a curl sign for upwelling.

- In conclusion, I did not get the impression that we can learn anything new about the pre-conditioning and causes of a Weddell polynya event from this suite of data. I note, that this example is quite late in the year from spring with applicable solar radiation being present and possibly reduced ice formation taking place in leads and polynyas - as compared to, e.g. July or August. I note further, that lateral water mass transport by eddies is not mentioned, even though it could well trigger similar T and S changes as those observed.

Lines 243-255 / Figure 8:

- I have the same comments with respect to technical aspects to Fig. 8 as for Fig. 7 - i.e. range of parameters, quality control, T at 0 m suggests permanent ice melt.

- Why is the decrease by 2 K surprizing?  Did you check whether your SIC-based delination method potentially simply failed here?

- I doubt that the salinity sensor at depth 220 m (panel f) provided reasonable values. Why is S measured there that stable compared to the depths above and below?

- The upwelling event starting at day -15 appears to be a weak one following the scale chosen in panel c).

- Line 248: "reach the deeper levels" –> There are oscillations at deeper level well before day -15.

- If that 3-day period indicated caused an upwelling, why don't we observed a T-increase and a S-increase in the layers closer to the surface above the CDW temperature maximum? One could argue that T at depth 170 m increased a bit during days -7, -8 compared to the period -15 through -9 but is this a signal of upwelling? In order to better delineate this it would be help full to obtain a better illustration what, in this case, the mean starting T and S values were to understand at which depth levels vertical water mass movement would cause which change in T and S. I don't find this overly conclusive. To my opinion, the phase from days -18 onwards is characterised by more or less intense ice formation with the concomitant variations in T and S at depth 170 m with an overall slight cooling and slight freshening at that depth. Associated variations in T are visible, also starting at day -18, at depths 220 m and 270 m; furthermore, also at larger depths there is a slow but noticable cooling, starting at around day -18, ending at day -10 and commencing again - depending on depth at day -6. I don't thing these events are overly well connected to wind speed curl signs or magnitudes.

- I agree with you that the event at days -2 / -1 is a nice example and well supports the notion that the upwelling conditions following the peak in divergence on day -2 can be well connected to the decrease in 0 m water temperature and also the deeper level decreases in T (down to 420 m depth). However, what I'd like to note is that decreases in T at depths 170 m, 220 m, 270 m and even at 320 m began at day -6 for which you don't offer an explanation even though this cooling might have been key for both an actual upwelling keeping the lead/polynya open and allowing for cooling of the surface

waters and the cold water pulse at larger depths on day -2.

- Finally, I note that according to the ASI-algorithm AMSR-E SIC maps at 6.25km grid resolution this event (the Oct. 9 2004 one) is one with a lot of smaller-scale openings to the south of Maud Rise which - to me - have more the characteristics of leads than of the classical Weddell Polynya observed, e.g. in 2016.

Figure 9 and its interpretation.

- I first focus on the time series without taking the SAR images into account. Yes, I agree the strong peak in divergent conditions at days -15/-14 coincides with a phase of considerable variation T and also S which, however lasts from day -15 through day -12; the lowest T and lowest S are actually observed on day -12; perhaps this could be explained with the time the cooled water requires to reach to the depths shown. But, there is similarly strong divergence event at day -27 which is not assoociated with such pronounced variations in S and T. In addition, there is another pronounced divergence event at day -9 which is associated with T and S maxima - diametrally different to the other two cases. There are other examples along the time series where curl sign and magnitude are not uniquely associated with variations / increases / decreases in T and S; hence I don't find this figure overly conclusive either. This is supported by the findings from Figure 8 where at these depths the correspondence between curl sign and magnitude was rather weak. I note that both depths are located below the depth of CDW temperature maximum in Fig. 6.

Now to the SAR images:

- Please provide the acquisition times for these images (well, you will have done so in the revised version in the context of section 2) and, in addition, mark there place on the time axis of the curl time series by a vertical black line.

- By the qualitative discussion provided you cannot give any statement about whether leads / openings seen in the three right SAR images are open or refrozen.

- In addition to the "0" indicating the polynya center I recommend to show the location of the mooring. If it is outside, then please state it clearly in the discussion of this figure.

- The fact that the SAR images supposedly provide radar backscatter ranging from 0 to -60 dB casts doubts upon having chosen the correct product.

- If you are after checking out whether leads and or the polynya was present I recommend to take a look at 3.125 km AMSR2 sea-ice concentration maps provided by the University of Hamburg: ftp://ftp-projects.cen.uni-hamburg.de/seaice/AMSR2/3.125km/

Line 281-283: "yet, our study ..." –> I cannot proof your statement wrong. But I am inclined to state that those "polynyas" identified by some of the papers mentioned were not the classical stype open Weddell Polynya but rather an agglomeration of leads. These also have the potential to reduce passive microwave SIC below the thresholds used - particularly because 100% thin ice - as is often observed in leads - is not seen as 100% SIC but considerably less than that (Ivanova et al., 2015, The Cryosphere). Also, I note that some of the authors mentioned here wrote about a "halo of low ice concentration" instead of a fully developed open ocean Weddell Polynya as observed in 2016/2017. I hence recommend strongly to re-phrase this statement.

Line 284-291: The way you presented the APP data set and its cloud screening as well as the way you described and applied the method did not convince me that your approach provides credible results. See my resepctive comments earlier.

Apart from that: The "Cape Darnley polynya" is not an open ocean polynya. This needs to be corrected in the manuscript. What IS an open ocean polynya is the Cosmonaut Sea polynya which actually popped up this past Austral winter as well as in 2016 (mid August).

Line 292-298: While I agree that the "Weddell Polynya" as you defined it in this paper (fully developed open ocean polynya but also an agglomeration of leads) is partly triggered by upwelling - aka - oceanographic conditions and partly by divergence - aka

- atmospheric conditions, I don't think that your paper hits the core. The debate is what triggers the fully developed open ocean polynya and not what causes a number of leads to open in the middle of the ice pack.

Line 299-301: I agree that high-resolution satellite observations are a crucial tool. But your study did not overly well demonstrate the importance of high-resolution data such as S-1 SAR. In addition, there are two sentinels carrying a SAR and the coverage is considerably better than you present here. In addition SAR IS operationally used by certain agencies / groups and institutions who are smart enough to handle the immense data amount associated with these high-resolution data. Finally, there is not just Sentinel-1a/b but there is PALSAR, TerraSAR-X, Tandem-L, COSMO-Skymed and RADARSAT / RCM SAR - all currently in orbit and functioning ... hence there is ample of such high-resolution data available - sitting in the archives to be used. Statements such as more sensors "onboard more satellites" should be very well thought about and backed-up by an appropriate analysis - an precondition which is not given by this paper.

######################################

Typos / editoral remarks:

General:

I find usage of "early warning system" not overly appropriate. There is not danger or damage or anything involved. It is too figurative in my eyes and I suggest to write something along the lines that you aim to better understand processes that precondition opening of the Weddell Sea polynya and that you are after to better predict its opening by means of a set of proxies derived from satellite remote sensing.

Line 7 and later in the manuscript, e.g. in section 2.3: "brightness temperature" is a term used in satellite microwave radiometry as well. Therefore I recommend to write "infrared brightness temperature" all the time in your manuscript you are referring to the AVHRR TIR data.

[Figure]

Line 5: "30 polynyas" –> "30 occurrences of the Weddell Sea polynya" or "30 polynya events".

Line 8: You use "area" two times in this sentence and it is fine to not mix it with "foot-print" which is often used to characterise the field of view satellite sensors have at the surface. So "along with a footprint at least larger than" should be replaced by "over an area larger than".

Line 26: "polynyas are holes formed in the winter pack" –> Is this the way WMO de-scribes what a polynya is? I suggest to revisit the definition and also make clear that we are not talking about true holes (like made by a needle) but that we are rather talking about an opening in sea-ice cover of a certain size, promoting water mass modification and/or sea-ice formation during the freezing season and melting during spring, being open water and/or covered by young and/or thin sea ice.

Lines 27++: Yes, I agree that some polynyas in fact have the potential to do so but not all of them, hence the formulation should be changed towards "can modify" or "could modify". The impacts you mention are determined by the local water masses.

Line 48: "low" –> "coarse"

Line 76: A reader knowing about the cloud coverage over Antarctic sea ice immediately ask herself / himself how many clear-sky images there would be within a typical winter month and whether the frequency if such images isn't too low to appropriately derive pre-conditioning information.

Line 77/78: "We only use the 2 AM ... year-round with dark images" –> I don't get this. First of all, I understood that you are not working with the reflectances but with infrared brightness temperatures (IRBT). Secondly, polar day commences Sep 22, a date which I assume is still within you search window, ... so there is no darkness anyways at the most southern end of your region of interest (ROI). Please clarify your writing.

Figure 4, caption, line 4: "Shown both" –> "Shown are both"

[Figure]

---

## Referee Comment (RC3) · Anonymous Referee #3 · 8 Oct 2020

The manuscript has serious weaknesses and is largely not reproducible. First and foremost, the aim of the study is unclear to me. Is it about improving remote sensing methodology or a forecasting system for the opening of the Weddell Polynya? Or both at the same time? The not very scientific approach and the sloppy writing is bothering (e.g. "debate is closed", "infrared out of fashion").

There is a lack of hypotheses, statistical tests for the significance and descriptions of the uncertainty. References are used in the wrong context and much of the existing literature is ignored. The description of the data is careless and incorrect in some places. If the goal were to analyze the AVHRR data using a new methodology, the

question of cloud cover would first have to be analyzed more thoroughly. It remains e.g. unclear whether passive microwaves and AVHRR data give consistent results. This would be the first step towards a suitable long-term study.

An investigation into the causes is difficult to do with observational data alone. The Weddell Polynya is a phenomenon in the coupled system of ocean-ice-atmosphere and is based on feedback effects and tides. Forcings such as fresh water fluxes through precipitation or melting of meteoric ice and the heat reservoir in the deep ocean play a role. Without a coupled model system, causal research is inadequate or must remain empirical. The empirical aspects of the study are however not solid because of the lack of hypotheses, statistical testing and significance.

However, the topic is very suitable for the journal and I suggest that the author should resubmit the work after a major revision. Because my concerns are about the main aim of the study I would encourage the author to withdraw the study and to resubmit it with better defined scopes, e.g. in two parts. First a validation of the AVHRR approach to detect the polynya, and a second part about the forecast method. The first part shall include a thorough error analysis about the influence of clouds. The second part should use advanced statistical methods.

References https://doi.org/10.1175/1520-0485(1999)029%3C1251:FAMOAP%3E2.0.CO;2 https://agupubs.onlinelibrary.wiley.com/doi/full/10.1002/2015GL064364 https://doi.org/10.1175/JCLI-D-16-0741.1 https://link.springer.com/article/10.1007/s10236-001-8173-5

---

## Author Comment (AC1) · 5 Feb 2021

We thank the reviewer for their comments. The role of the reviewer has been duly acknowledged with the addition of this sentence in the acknowledgement section:

"We also thank the two anonymous reviewers 1 and 3 and Stephan Kern (reviewer 2) for their comments that greatly improved the clarity and quality of this manuscript"

General comments: 1. The summary of satellite observations for sea ice monitoring needs improvement. A more thorough reading of the relevant references is needed.

» We thank the reviewer for the more specific comments they made further down in

their review. All changes to the introduction that were suggested have been made.

2. The method is not clearly described in the manuscript. A flowchart of the "early warning system" could be helpful. For example, how the polynya-prone area is determined and how it varies with years? In Section 3.1, when determining the polynya dates, is the minimum SIC area-wise (the polynya-prone area)? If yes, is the polynya-prone area fixed? How large? How many pixels are we talking about? Such questions pop when we start to read section 3.1, however only find it till the end (in the caption of Figure 3). In section 3.2, why plotting the data on the 15-day minimum and maximum axes? It is not clear that, what are the final criteria for the early detection until I read the manuscript a couple times.

» We agree with the reviewer that the methods were not descriptive enough. We have heavily rewritten the manuscript to add clarity. Furthermore, we have now made our codes freely available on Github, as described in the Code Availability section. We added a flow chart as the reviewer suggested (new Figure 6). We reformulated the description of the polynya-prone region as early as section 2 to answer most of the reviewer's questions: the coordinates are fixed and based on existing literature. We also provide the area and corresponding number of grid cells. Then we added a sentence at the beginning of 3.1 to remind the reader of the main characteristics of this region. Section 3.2 has been clarified and figure 4 has been changed. The final criteria are clearly spelled out in section 3.2, in the conclusions, and indicated on the flow chart.

3. It is a bit difficult to follow the flow of the manuscript, especially for the method. Section 3.1 and 3.2 are part of the method however presented in the "Results and Discussion" section. It could be easier to read if restructuring the "Method" and "Results and Discussion" sections, with "the warning system" (and identifying criteria) in "Method" and "cause of the opening" in "Results and Discussion".

» In response to the other reviewers' comments, we restructured differently from what the reviewer here suggests. The detection method is the main result of this paper; the

cause of opening, an extra result to discuss the usability of infrared data. Hence we dramatically reduced section 3.3 and turned it into section 4: Discussion.

Specific comments: P1L1: "a crucial information" to "crucial"

» Changed.

P2L45-46: "sea ice concentration from passive radiometers or spectrometers (Spreen et al., 2008), mostly in the microwave region..." I don't recall any spectrometers used for sea ice concentration retrieval. The most commonly used data to derive sea ice concentration is microwave radiometer data, because the sensor (i.e. microwave radiometer) can work independence of light and penetrate clouds. There is no need to mention spectrometers. It can be rewritten to sth like "...sea ice concentration from microwave radiometers (Spreen et al., 2008)."

» That was an unfortunate typo. Thank you for noticing it, the text has been changed as suggested.

P2L46-52: The authors give a list of references to present the low and high resolution applications. It is true that SAR has much higher spatial resolution than microwave radiometer, BUT both are microwave sensors. It is not appropriate to use "microwave products" to refer to the radiometer/scatterometer-based products. On the other hand, it is common to use microwave radiometers for sea ice drift tracking, sea ice classification and thickness retrieval (e.g., SMOS/SMAP). In Korosov et al., 2018, the microwave radiometers were firstly used to derive sea ice drift information, on the basis of which sea ice age was tracked. These sentences should be rephrased. Please also check the use of "microwave" (when referring to microwave radiometer), in the whole manuscript.

» "Microwave" has been constantly changed to "passive microwave remote sensing" to increase clarity. The reference to Korosov et al. 2018 has been removed and replaced with a more relevant citation. Throughout the manuscript, "microwave" has been removed and replaced with ad-hoc, less ambiguous formulations.

P2L52-53: Microwave radiometers have been used to monitor sea ice since late 1970s,they are not recent sensors compared to infrared sensors.

» This sentence has been rephrased to "Moreover, SAR is a comparatively recent technology for sea ice observation", i.e. the reference to passive microwave has been removed.

P3L59: "determine the mechanisms" to "understand the mechanisms"

» We disagree with the reviewer's suggestion here – in this paper, we genuinely solely aim to determine, not build an understanding of the dynamics.

P3L69: "reanalysis"

» This sentence has been rephrased to: "in-situ hydrographic and atmospheric reanalysis data"

P3L72-74: since these two datasets have different resolution, please specify the resolution respectively.

» This part has been changed. All dataset used have their resolution specified.

P6L128-129: "using the traditional criterion sea ice minimum...", do you mean sea ice concentration minimum? It is not clear here how the "sea ice minimum" is determined? If this is the minimum concentration of the polynya-prone area, a description of how the polynya area is determined and how large is the area is needed.

» This comment is the same as major comment 2. As we explained before, both section 2 and section 3.1 have been rephrased to answer these question.

P7L140: "yield" to "yields"

» Corrected

P7L143: "then a third opens...the largest one splits into two". It is confusing here, which one is the largest one?

» This paragraph has been removed

P7L147-148: "thirty" to "30";

» Changed.

P7L166: 3443 days? For how many years? In the next paragraph, the number of days is replaced with the number of events? Why?

» This paragraph has been rephrased for clarity.

P7L160-171: These two paragraphs basically explains that threshold for single band does not work well for early detection but the three bands together do. A summary of the final criteria and the reason is needed.

» This section has been heavily rephrased. A summary has been added as suggested. The final criteria are also mentioned on the flowchart that the reviewer suggested we add (new Figure 6).

P12L220: which three events do you mean? Do you mean the average temperature in the 15 days, or when the opening starts?

» Rephrased to "three polynya events are colder than or around -10 °C over the entire 15-day period that preceeds the polynya opening".

In Figure A5, there are more cases with 2 m air temperature of -10 degree by the time of opening.

» This comment from the reviewer is related to the confusing formulation of that sentence, which has now been changed in response to the previous comment.

P13L235: I don't find temperature and salinity at 480 m depth oscillate with a near 12hour frequency, at least not from -30 days to -13 days. Only smaller oscillations are found during 30-20 days before the event.

» This sentence has been removed in response to reviewer 3.

P13L250: Again, I don't find increase of temperature at depth. It is too small to notice. The brightness temperature increases following the dip of T45 however only for one day.

» This sentence as well has been removed in response to reviewer 3.

P7L257-258: I would suggest delete the two question mark and words in parentheses.

» This paragraph has been rephrased.

P15L265: "rejected"

» Changed.

P17L300: "Sentinel" to "Sentinel-1", the revisit time for the twin Sentinel-1 satellites is 6 days not 4 days

» Changed to Sentinel-1. We agree with the reviewer that the revisit time is 6 days over the same satellite orbit. However, from different orbits the same place can be monitored more often, especially at higher latitudes (as visible on Fig 9).

---

## Author Comment (AC2) · 5 Feb 2021

We thank the reviewer for their comments. The role of the reviewer has been duly acknowledged with the addition of this sentence in the acknowledgement section, which explicitly names the reviewer as was agreed during an email conversation with the reviewer:

"We also thank the two anonymous reviewers 1 and 3 and Stephan Kern (reviewer 2) for their comments that greatly improved the clarity and quality of this manuscript"

General comments (GC):

GC1: I have a general concern with the APP-X data used and in particular with their quality. Since this is a key data set of this paper it deserves a much more detailed consideration, especially with respect to clouds which masking is, to my opinion, not convincingly illustrated. Here the manuscript requires some additional figures / illustrations as well as some re-writing

» We would like to start by confirming that the dataset we refer to as "APP" is indeed APP, not APP-x which would have too coarse a resolution for our study. We did nevertheless add an entire appendix dedicated to validating the cloud masking (based on published literature) against the MODIS/Aqua cloud mask product (MYD35_L2), as well as extra figures to investigate the cloud conditions in the days leading to a polynya event compared to a "normal" year.

GC2: I also have a general concern with section 2.3 which I find too generic. A scientist wishing to repeat you study would be lost because the description of the methodology is not adequately detailed. Here the manuscript requires some re-writing. In addition, I have the impression that choosing a different SIC algorithm would result in different "polynya events"

» A scientist wishing to repeat our study will now find all our codes freely available on Github, as we detail in the Code Availability section. Following a suggestion from reviewer 1, we also added a detailed flow chart to better describe our methodology. Overall, the paper has also been heavily rewritten to improve its clarity. Regarding the SIC algorithm, we would like to insist on a message we wrote repeatedly in the paper: although the only Weddell Polynya events that are mentioned in most of the literature (see a summary in Campbell et al. 2019, and references therein) are those of 1974-1976 and 2016-2017, we find other events, the same as found by authors before us that we cite in section 3.1. The only difference between these authors, which is most likely caused by different algorithms and/or data products, is in the area of the polynya and its duration. Consequently, and as we already indicate in section 3.1, the polynya events are not SIC-algorithm dependent.

GC3: I have a general concern about the applicability of the various infrared TB parameters. The caveats of the parameters used to delineate polynya conditions or conditions leading to the formation of a polynya are not sufficiently well laid out. Taking the statistics of IR TBs over an about 500 000 sq km large region as a measure to better delineate the presence (or to-bt-presence) of a polynya of a size 1% of that area appears not to be well motivated.

» We do not understand what the reviewer suggests here. We motivate throughout the manuscript that our reason for using area-wide statistics is that the overall, ideal aim of the project is to generate algorithms that would be automatised. Working only with the exact area where the polynya opens is unrealistically optimistic. Furthermore, as is summarised in e.g. Swart et al. 2018 (cited in our manuscript), the 2017 polynya reached a maximum area of 295 000 km2, i.e. covered half of our "polynya-prone" area. As has been found by many authors, especially modellers (e.g. Dufour et al. 2017 and references therein), such large area can be reached in part by pre-conditioning of the ocean over an even larger area. Hence, it is sensible to work with area-wide statistics.

GC4: I have a general concern about the interpretation of the various infrared TB parameters - particularly in the context of figures 7 through 9 when it comes to the examples. The interpretation of observed changes in these IR TB parameters appears to be biased towards oceanic and sea-ice conditions, leaving out the direct influence of the atmosphere on the sea ice / snow surface. By the same token, a critical discussion of the usefulness of the wind data for the curl computations is missing.

» Following reviewer 3's comments, most of this discussion has been removed. Consequently, what is left is now balanced between atmosphere (air temperature and wind) and ocean (mooring analysis). Regarding the wind curl, we would be grateful if the reviewer could indicate more specifically what is missing, as we consider that we already address this comment in the last paragraph of section 2.3.

Specific comments

Lines 17-21: I doubt that motivating the importance of the Weddell Sea polynya, which is a winter phenomenon, with commercial exploitation of ice-infested waters is a correct thing to do for the Antarctic. Most tourism is in summer; fishing occurs outside the sea-ice cover; transportation does not happen, and exploration is limited thanks to the Antarctic Treaty. Hence, it is rather an interesting bio-geo-physical event with highly interesting ocean-ice-atmosphere interactions than anything else.

» In that sentence, we are not motivating at all the Weddell Sea polynya. We are talking in general about "global changes in sea ice cover", and later about "sea ice opening". Consequently, we did not change this sentence.

- Line 47: Satellite passive microwave data are used to derive sea-ice motion. Based on the sea-ice motion sea-ice age is derived; while Korosov et al. (2018) worked on along-needed adjustment of the motion-to-age conversion the original work dates back to 2003 or 2007, I guess. Maslanik or Tschdui should be the main authors, citations you can find in the paper by Korosov et al. (2018).

» We thank the reviewer for this rectification. We changed the reference to Maslanik et al. (2007)

- Line 48: The algorithm proposed by Spreen et al. (2008) was based on AMSR-E data, the predecessor of AMSR2. That satellite did not allow for the stated fine resolution of about 3 km. One of the few citations pointing towards that there is a version of the same algorithm applied to AMSR2 data which allows for about 3 km grid resolution would be the one of Beitsch et al. (2014) in the journal "Remote Sensing", vol. 6.

» We thank the reviewer for this comment. We had not seen that our formulation, putting on the same line the reference to Spreen et al. (2008) and AMSR-2 frequency, was indeed confusing. We instead give as exemplary frequency the highest of AMSR-E.

Lines 49-51: If you keep them, then all these citations require an "e.g. in front of the

author(s) because there is a full suite of papers into these directions ... except the one for melt ponds; that one I suggest to delete because if you read that carefully you will figure out that the way they tried to infer melt ponds from HH-Pol wide swath SAR imagery did not work out well. While there have been many studies involving SAR and melt ponds I doubt THE solution has been found here and therefore it is for sure NOT the norm to use SAR for melt pond detection. Furthermore SAR and sea-ice thickness retrievals is also something I would not put under the "norm" to use SAR imagery. The range of maximum sea-ice thickness values to be retrieved using SAR goes from 0.1 m to 2 meters; this is a field which has not yet been explored enough and we are also not yet there - despite what machine learning and other similar approaches attempt to sell us. SAR imagery is extremely ambiguous and not really suited for sea-ice thickness retrieval.

» We agree with the reviewer that there are more citations than just the ones we give, but we like our papers concise. We followed their advice and added "e.g." to most references. We did not see how to elegantly indicate that melt pond detection did not work as well as the authors would have liked, but still want to acknowledge that it is an area of research, so we kept this reference. We changed the formulation from "the norm" to "common". We agree with the reviewers that SAR-based retrievals for sea ice thickness are far from mature, but as reviewer 3 insisted that we even talk about modelling, a reader would be surprised that we do not at least mention SAR applications.

- Line 52: "microwave and SAR" –> SAR is microwave. "comparatively recent sensors" mentioned together with a 20+ years old citation does neither look overly smart nor is it correct, because microwave sensors date back to 1972/1973 with ESMR being the first one to allow to observe sea ice independent of clouds and daylight - being the sensor to allow detection of the Weddell Polynya for the first time.

» These two comments have already been addressed in response to a similar comment by reviewer 1. We now distinguish the two by referring to "passive microwave remote

sensing".

» Twice in this document the reviewer uses the term "smart" to hint that we are anything but. We fear that the reviewer is not aware that "smart" is inappropriate in a review as it enters the realm of personal comments, and as such should be avoided. We suggest the more neutral "not relevant". Either way, we rephrased this sentence to make it clear that it is SAR applications that are recent, and that 20 years is comparatively short when studying the climate system.

- Line 55: "hence has fallen ..." –> Same comment as above in terms of 20+ years old citation plus: There has been quite some work with spaceborne infrared sensors during the past two decades. MODIS, for instance, has been used heavily for polynya detection in both hemispheres. It has been used for fast ice detection in the Antarctic. AVHRR and nowadays VIIRS has been used for sea-ice cover retrieval and, more importantly thin ice thickness retrieval quite extensively as you may note from papers by Key et al., 2016, The AVHRR polar pathfinder climate data records, Remote Sensing,8 or Mäkynen and Karvonen, 2017, MODIS sea-ice thickness and open water-sea ice charts over the Barents and Kara Seas for development and validation of sea-ice products from microwave sensor data, Remote Sensing, 9. It might therefore make a lot of sense to better embed you work into the current state-of-the-art after you have undertaken a little literature review.

» We removed this sentence in response to comments by the other reviewers.

Introduction: I am sure a reader would appreciate a short paragraph in which you very briefly describe which data sets you applied to achieve which findings so that the list of data which follows falls on well-prepared ground.

» We have to disagree; this would make the paper highly repetitive, as the data are described one paragraph later. As a compromise, we now name the datasets in the final paragraph on the introduction (paragraph that has been rephrased in response to reviewer 3).

Section 2.1:

- I suggest to provide a table in which you summarize which data you use at which spatio-temporal sampling for which time period. Also the grid information (i.e. which grid is used) should be included.

» We understand the reviewer's suggestion but could not find a satisfactory way to summarise all relevant information in a table with common headers, as each product has its own specific characteristics that need mentioning. We instead expanded the description of each product in response to the reviewer's comments below. We do not understand what the reviewer means with "grid", as we already specify the horizontal and temporal resolutions.

Line 67 and line 73 contain contradicting time information: 1980 vs. 1978.

» We corrected this mismatch, thank you for pointing it out.

- There is a more recent version of the APP data set called APP-X. Did you use the latter one?

» No, we used APP, which has a more relevant resolution for this study.

- I note that you provide a doi for the ERA5 data but not for the other data sets? Aren't there doi's for other data sets as well?

» We now provide all the dois already in this section, instead of the Data Availability section solely.

- I note that your description of the data is quite light and poses questions.

» We added a sentence at the beginning of this section to motivate our conciseness and encourage the reader to consult the respective data description papers that we cite.

a) Which algorithm is the long-term sea-ice concentration data set from NSIDC based

upon. If it is the NASA-Team algorithm then you need to be aware of the events of substantial underestimation of the sea-ice concentration due to snow metamorphism which this algorithm is subject to in late winter / spring. Other algorithms and products, i.e. the Comiso Bootstrap v3 or the OSI-450 data set do not have these problems; particularly the latter one, which is a CDR, appears to be most suited for your purpose.

» It is not the NASA-Team algorithm, it is the Comiso Bootstrap v3. We have added this precision to the text.

b) Which resolution has the "product from the University of Bremen"? On which algorithm is it based? Over which time period did you use these data? Did you consider to use the 3.125 km product offered by the University of Hamburg (now from AWI) being available since 2012 based on AMSR2?

» We no longer use this product, as the section for which it was needed has been removed.

c) "Hydrographic data" is quite generic. Which data measured how at which depths over which time periods with which instruments?

» We are glad you asked. We added two (long) sentences to answer the reviewer's comment.

d) Even though you used SAR only for qualitative purposes it would be important to learn how many you used, which time-differences these have with the other remote sensing data, and so forth. If it is just a dozen or so you can simply provide a table with overpass time, time difference to whatever other data, et cet. In addition "backscatter information" is also very generic. What is the purpose and what were you after in particular. One sentence stating that you were trying to find open water / new ice signatures in C-Band HH backscatter images would possibly be enough.

» As suggested, we added the dates, time, and difference with the APP product acquisition. We also added the suggested sentence.

e) The AVHRR data set used ... is this an FCDR or CDR? Is its inter-sensor bias corrected? I am asking because it is known that the various individual satellites hosting AVHRR sensors have been prone to serious drift in their orbits, shifting local overpass times by up to several hours.

» It is an FCDR, as specified in the Key et al. (2019) dataset reference that we cite. The same reference explains (we here paraphrase their page 3) that the fact that APP is made of composites several hours around the target time rather than single images mitigates the drift issue.

f) Line 80 "For validation" –> What did you evaluate? Please add.

» Sentence clarified.

Section 2.2:

I suggest to considerably expand this section. The illustration of the cloud mask is not convincing. Since the detection of the polynya and/or its preconditioning conditions is among the key elements of this paper and since these infrared data are the key dataset to be used this section deserves a lot more attention to ensure that any conclusions taken are not simply taken because of either an insufficient cloud mask or a drift in the cloud mask capabilities to mask out clouds or a drift in the used APP-X product.

» As we have already indicated in previous comments by the reviewer, we are not using the APP-x product. We are using the APP product. This product is corrected for drift. Following the reviewer's suggestion, and also in response to reviewer 1, we added an appendix in which we validate the cloud mask (see also comment below).

One way to check the cloud mask would be to show coincident high-resolution MODIS and/or Landsat images. It could be sufficient to have the reflectances only because structures and textures evident in the reflectance and the infrared images might be similar. In any case would it be much better to actually see several cases where your approach correctly allows to delineate the surface temperature - i.e. by showing the

temperature gradient across the ice edge or within a polynya area. Currently, this section is not convincing - neither about the usefulness of the APP-X IRBT data nor about the cloud masking scheme.

» First, once again, we are not using the APP-x data. As we wrote, we use the APP data as described in Key et al. (2019). We do thank the reviewer for their suggestion of investigating coincident MODIS images. We hence discovered a MODIS-based cloud mask product, peer-reviewed, and highly recommended by our cloud-physics colleagues. We use this product (MYD35_L2) to validate the usage of the Yamanouchi et al. (1987), Vincent et al. (2008) and Vincent (2018) criteria for cloud masking. We did find during this validation exercise that the Yamanouchi et al. (1987) criterion on T34 was not restrictive enough, and hence modified it in section 2.2 and for the rest of our analysis, as we describe in the text.

In this context, it might make sense to look at the recent paper of Vincent et al. from 2019 in Remote Sensing, 11, "The case for a single channel composite Arctic SST algorithm".

» We thank the reviewer for their suggestion. We added instead a reference to Vincent (2018), on which Vincent et al. (2019) was built, and which is more directly relevant as it contained a specific cloud mask criterion. We added this criterion to our cloud mask (included in the tests of appendix A).

Section 2.3:

- Line 105: Please specify in your text when and why you use a per-pixel SIC threshold or an area-average threshold. It is not clear why both are used and/or required.

» We rephrased to clarify that we here use methods published by past authors: some use a per-pixel SIC threshold, some use an area-average threshold.

- Line 107: "we tested .." –> Very generic description. Please provide more details in your text. How did you visually assess each option? Against which information did you

do this assessment? Over which period and with which frequency of the input data did you carry out this testing?

» Text has been modified to be more specific.

- Line 108: "contiguous pixels" –> How did you assure / identify whether pixels are contiguous or not? Please provide more details in your text.

» We reformulated this sentence to clarify that we first perform a contour detection. We also provide the code.

- "pixel area" –> What did you use as the pixel area. If you have used data on a polar-stereographic projection, which I assume you did, then you should have taken the grid-cell area of each individual grid cell from a separate data file provided by NSIDC. This needs to be clarified.

» The latitude and longitude grids are not provided by NSIDC and instead have to be computed by the user. We provide ours at the same Github link as the rest of our codes, which we already mentioned earlier.

- Line 109: "60% for most of the study" –> What are the incidences where you needed to use a different threshold and why? Please be more specific in your text.

» We modified this sentence. We consistently use 60% in this study.

- Please provide information about the time period per year you use. Currently one can think that you try to do the analysis for the entire year (which of course is not logical because of the lack of sea ice in the polynya-prone region for a number of months. But you don't tell the reader your time period of interest.

» We do use the whole year in most instances as the seasonal cycle is relevant for our criteria (see the new figure 5 and the new supplementary figure B1).

- Line 105-110: For what is this data set created? What is the purpose and how is it connected to the other parts of the method?

» We added a sentence at the end of this paragraph to link it to the rest of the study. In response to the reviewer's subsequent comment, we also added to that new sentence that we detected more than 20 polynya events with this method, and that the reader will know more in section 3.1 (two paragraphs to wait).

- Line 113: "For robustness though ..." –> It is not clear from your text why this step is needed to check the robustness. It is also not clear what you mean by brightness temperature composites. This requires further clarification if kept.

» Sentence removed as suggested.

- Lines 115/116: The "anomalies" you are writing about, did you compute this separately for all parameters, i.e. did you compute daily climatological values for median, minimum and maximum IR TB?

» We agree with the reviewer that this sentence was confusing. We moved the anomaly description upward.

- Line 116: "over the 40 years" –> It is 37 years, isn't it?

» It is 38, 1982-2019 included. We modified throughout the text for consistency and clarified the difference between the APP time period and the SIC time period.

- Line 117: "using only the years ..." –> Is this surprizing? Possibly not because if I understood your introduction / motivation correctly it was 2016 for the first time that the Weddell Polynya opened again within the considered 1982-2018 period ... hence the majority of the data are not including polynya conditions anyways.

» As we show in section 3.1 and as already discussed in this review response, there were more events than just the "famous" 2016 one. We did remove this confusing sentence.

- Line 119: "over each polynya" –> I though you are only targeting the Weddell Polynya so which other polynyas referred to by "each" are you investigating here? It is not clear

from your writing why there should be several polynyas.

» We understand the reviewer's confusion. As already discussed throughout this docu-
ment, there are more than 20 polynya events in the time series, over the same region.
This is what "each polynya" means. We reformulated.

- Line 125: Please provide a reference for this interplay between convergent situations
and upwelling.

» We added a reference to our favourite 1st year undergrad oceanography textbook.

Section 3.1:

Figure 2 / Line 130: What motivates using a mean SIC threshold of 92%?

» We have rephrased so that it is clearer that we did not arbitrarily choose 92% but
simply used a criterion from a past, peer-reviewed study by a different group.

Line 137: Which "other methods" are you referring to here?

» We rephrased to make this sentence clearer.

Line 139: "visual validation" –> I would certainly not term this "validation"; I'd rather
say you did a plausibility check. How did you assess the 60% criterion quantitatively?
Is there any quantitative information you can give young students and/or scientists at
hand who want to repeat you study?

» This joins the earlier observation by the reviewer on the disappointing results of
machine learning methods applied to sea ice remote sensing: if there were an easy
way to quantify what the trained human eye sees, we would not need human analysts
anymore. And having been staring at sea ice maps looking for polynyas from the
early days of her PhD, C.H. is trained. We did remove the term "validation" from the
sentence.

Table A1: Did you check your analysis with respect to the credibility of the polynya

events detected? Looking at Table A1 I am concerned with cases where the polynya (according to the 60% criterion) lasted just 1 or 2 days but occupied a comparably vast area of nearly 10000 sq km. How realistic do you consider such a rapid opening and closing of the polynya?

» We refer the reviewer to the detailed work of our PhD student, Martin Mohrmann, recently submitted to The Cryosphere (reference in our text), which shows that as surprising as it sounds, it is a common phenomenon.

Line 151: Following up with my earlier comment about which sea-ice concentration product you used I recommend to repeat the same analysis (and in particular the numbers shown in Table A1) using a different algorithm, e.g. OSI-450 or Comiso Bootstrap v3.1. This way you would be able to considerably enhance the credibility of your findings and eventually eliminate the cases of an overly rapid opening / closing of polynyas of considerable size.

» We decided to not show these results in the paper to not make it unnecessarily heavy, but we did reproduce these results using other sea ice products (like that of the University of Bremen, and OSI-450) as well as sea ice thickness for the most recent years. As specified earlier: we do use the Comiso Bootstrap v3.1 product. We have clarified this in the text on several occasions.

Section 3.2:

» This section has been heavily rewritten, in response to the (justified) comments by all reviewers that it was really hard to follow and understand. The figures have also been modified to make our point clearer. Consequently, to avoid making this document unnecessarily long, we answer only the one comment from the reviewer that is still relevant:

Lines 189/190: It is not unlikely that there have already been leads in the pp-region before the Polynya events, isn't it? Regarding the T45 threshold I again refer to the

more recent publication by Vincent et al. I mentioned earlier in my review.

» It is not unlikely indeed. We added a few words to specify this. As interesting a read as this paper was, we did not see its relevance (aside from confirming that BT11-12, as it is called there, cannot be used for absolute SST determination).

Section 3.3 (now renamed Section 4)

» This section has been reduced as suggested by reviewer 3. Consequently, comments related to lines 232-255 or to Figures 7 and 8 are no longer relevant and will not be shown here.

Line 205: "the oscillation in (IR) TB suggests an upwelling of warm water" –> If I understood you correctly, then the examples of the oscillating IR TBs shown in Fig. 5 and inthe respective figures in the Appendix are based on observations of the polynya-prone region (PPR) shown in Figure 3, extending south-north over 8 degrees latitude which is480 nautical miles or about 800 km. Hence the oscillations shown reflect the conditions of a synoptic-scale region. It could (...) be that the oscillations and variations in IR TB are caused by ocean upwelling. But wouldn't this provide that a substantial amount of the sea ice is i) bare = not snow-covered and ii) sufficiently thin so that a 1-day long up-welling event manifests itself in a change in the median or maximum IR TB of the PPR by up to 7 Kelvin? What is the typical temperature variation immediately beneath the ice bottom / at the water surface in the leads due to an upwelling event? I'd expect it is an order of magnitude smaller. What - to my opinion - is missing in your initial scientific reasoning here is the immediate impact of the atmosphere. Advection of warm or cold air can very well cause surface temperature changes of the observed magnitude within the PPR. The IR TB measured is direct measure of the surface temperature. To my opinion, the size of the PPR in combination with the magnitude of the observed IR TB changes and the fact that maximum IR TBs are often de-coupled from the median IRTB (which is fine as mild air might prevail in the north and cold air in the south of the PPR), confirms that a direct atmospheric influence is much more

likely as a cause for the oscillations than oceanic upwelling. This is also supported by the knowledge that surface temperature changes caused by ocean upwelling require substantially more time than surface temperature changes caused directly by the atmosphere.

» If we understand correctly the reviewer's comment, they would like us to acknowledge the role of the air temperature not only in its dedicated paragraph, but also in the paragraph they here highlight. We added a sentence that summarises the reviewer's point, along with a reference to the Francis et al. (2020) paper that describes this exact phenomenon in relation to the 2017 polynya.

An issue being connected to this is that fact that we don't know how accurate the IRTBs are and how often cloud artifacts disturb the observations. IR TB artifacts due to clouds can create both comparably warm or cold signatures, depending on whether a low-altitude warm cloud or a high-altitude cirrus cloud was not adequately masked out. Furthermore, Figure 5 and the resepctive figures in the Appendix lack information about how many of the pixels of the PPR were actually cloud-free during the 14-dayperiods considered. My assumption would be that the number of cloud-free grid cells is considerably larger for the comparably cold maximum IR TB cases shown. In other words, the statistics of every data point shown in these figures potentially varies from day to day.

» We thank the reviewer for this suggestion. We added two columns to supplementary table B1 to indicate the median and minimum-maximum range of cloud-free pixels over the 15 day period that precedes each polynya event, along with supplementary Figure B1 that shows the seasonal cycle and range in cloudiness, demonstrating that the polynya years are not unusual. Although the number of cloud free pixels does vary from day to day, we do not understand from the first sentences of this explanation by the reviewer how this would impact the maximum IR TB value.

Figure A4 caption: I suggest to add the information that red (=positive) likely denotes

divergent - aka - ice cover opens mechanically while blue (=negative) likely denotes upwelling - aka - ice melt from below, or if there is sufficiently open water present already also lateral melt.

» We added this information to the caption.

Line 212: "it can open the ice within 12 hours" –> Lets take this as a useful number for the moment. In this case, within one day, divergent wind conditions could result in a drastic increase in the observed IR TB because an area covered completely by sea ice of a surface temperature of -20degC might change in an area covered by 95% sea ice and 5% open water of -2degC, which would result in an area-average surface temperature of -19.1degC, an increase by about 1K. It is straight-forward to estimate other surface temperature changes as a function of open water created. Note that this simple consideration applies to a case where the open water does not freeze over again immediately.

» We agree with the reviewer's reflexion. We would like to point out that lack most of the parameters required to quantify the change in surface temperature though. We added this precision to the text.

Lines 214-221 / Fig. A5. This paragraph and interpretation of the results requires some rewriting. - If I follow the considerations in the various Nandan et al. papers then I am inclined to say that surface melt of the snow cover would occur at 0degC most of the time – unless we talk about thin ice with a thin snow cover - which is something you don't know. Any thicker (above 6-8 cm) snow would possibly have zero surface salinity. - I agree that the sea ice would not start melting at 0degC but already at a lower air(surface) temperature. However, the majority of the Antarctic sea-ice cover is snow covered. - I am not sure what the intention of this paragraph is. Is it trying to use ONLY the air-temperature information to find a signal which could be interpreted as a pre-conditioning for polynya formation? This should be made clear in its first sentence.

» We clarified the purpose of our paragraph by adding a new first sentence. We have

added an extra sentence regarding the lack of information regarding the thickness of the snow layer.

- "The main caveat ... is colder" –> This reads as if the 2m air temperature is always colder than the surface temperature. I'd say the opposite is the case for snow covered sea ice - unless the surface was warmed by the mild air of a warm sector of a cyclone and now a cold front brings cold air advection. Hence your statement as given might only be valid for young and thin bare ice or for the special case just described and needs to be re-phrased.

» We did not write that it is always colder; we wrote that it "probably" is colder. We nevertheless rephrased as "it may be colder".

Figure 6: It there the possibility to show a similar plot for the salinity or at least an illustration of the typical vertical salinity profile during winter? This would aid in a better understanding that oceanic convection events would cause a decrease in T AND S at depth.

» The reviewer seems to have missed the reference to the supplementary Figure B4 that shows exactly this. We have increased the number of times we refer to this salinity figure to make it more obvious.

Line 229: "luckily, there were Sentinel-1 images" –> This information is out of context here because a SAR image cannot replace surface temperature or salinity measurements. I suggest to delete this information here and put it somewhere else at a more appropriate place.

» We removed this part of the sentence.

Figure 9 and its interpretation.- I first focus on the time series without taking the SAR images into account. Yes, Iagree the strong peak in divergent conditions at days -15/-14 coincides with a phaseof considerable variation T and also S which, however lasts from day -15 through day-12; the lowest T and lowest S are actually observed on day

-12; perhaps this could be explained with the time the cooled water requires to reach to the depths shown. But, there is similarly strong divergence event at day -27 which is not assoociated with such pronounced variations in S and T. In addition, there is another pronounced divergence event at day -9 which is associated with T and S maxima - diametrally different to the other two cases. There are other examples along the time series where curl sign and magnitude are not uniquely associated with variations / increases / decreases in Tand S; hence I don't find this figure overly conclusive either. This is supported by the findings from Figure 8 where at these depths the correspondence between curl sign and magnitude was rather weak. I note that both depths are located below the depth of CDW temperature maximum in Fig. 6.

» We have reformulated the description of this figure and made it clearer that the results from the mooring analysis are not really conclusive.

Now to the SAR images:- Please provide the acquisition times for these images (well, you will have done so in the revised version in the context of section 2) and, in addition, mark there place on the time axis of the curl time series by a vertical black line.

» We have added the acquisition times in section 2 already in response to an earlier comment from the reviewer. The times are already marked with a black arrow, and time periods refer to with grey boxes. We tried the reviewer's suggestion of extra vertical axes, but it made the figure too heavy and quite confusing.

- By the qualitative discussion provided you cannot give any statement about whether leads / openings seen in the three right SAR images are open or refrozen.

» We do not understand the reviewer's comment; open leads have a different backscatter from refrozen ones, so different that they are visible with the naked eye on such images.

- In addition to the "0" indicating the polynya center I recommend to show the location of the mooring. If it is outside, then please state it clearly in the discussion of this figure.

» The mooring is outside of the region shown on the SAR images. We added this precision to the text along with a discussion of how comparable the two are.

- The fact that the SAR images supposedly provide radar backscatter ranging from 0to -60 dB casts doubts upon having chosen the correct product.

» We do not understand what the reviewer finds suspicious about these values.

- If you are after checking out whether leads and or the polynya was present I recommend to take a look at 3.125 km AMSR2 sea-ice concentration maps provided by the University of Hamburg: ftp://ftp-projects.cen.uni-hamburg.de/seaice/AMSR2/3.125km/

» We understand that the reviewer is proud of the product he generated but do not understand why on several occasions in this document he suggested it for applications where it is not relevant. Here for example, using this product would force us to go back down to a resolution similar to that of APP, even though we use SAR specifically for its very high resolution.

Conclusions

Line 281-283: "yet, our study ..." –> I cannot proof your statement wrong. But I am inclined to state that those "polynyas" identified by some of the papers mentioned were not the classical stype open Weddell Polynya but rather an agglomeration of leads. These also have the potential to reduce passive microwave SIC below the thresholds used - particularly because 100% thin ice - as is often observed in leads - is not seen as 100% SIC but considerably less than that (Ivanova et al., 2015, The Cryosphere).Also, I note that some of the authors mentioned here wrote about a "halo of low ice concentration" instead of a fully developed open ocean Weddell Polynya as observed in 2016/2017. I hence recommend strongly to re-phrase this statement.

» We rephrased.

Line 284-291: The way you presented the APP data set and its cloud screening as well as the way you described and applied the method did not convince me that your

approach provides credible results. See my resepctive comments earlier.

» This sentence, along with the rest of the methods (including the cloud screening), has been rephrased already in response to other comments

Apart from that: The "Cape Darnley polynya" is not an open ocean polynya. This needs to be corrected in the manuscript. What IS an open ocean polynya is the Cosmonaut Sea polynya which actually popped up this past Austral winter as well as in 2016 (mid August).

» We removed the reference to the Cape Darnley Polynya.

Line 292-298: While I agree that the "Weddell Polynya" as you defined it in this paper (fully developed open ocean polynya but also an agglomeration of leads) is partly triggered by upwelling - aka - oceanographic conditions and partly by divergence – aka - atmospheric conditions, I don't think that your paper hits the core. The debate is what triggers the fully developed open ocean polynya and not what causes a number of leads to open in the middle of the ice pack.

» We tuned down these sentences.

Line 299-301: I agree that high-resolution satellite observations are a crucial tool. But your study did not overly well demonstrate the importance of high-resolution data such as S-1 SAR. In addition, there are two sentinels carrying a SAR and the coverage is considerably better than you present here. In addition SAR IS operationally used by certain agencies / groups and institutions who are smart enough to handle the immense data amount associated with these high-resolution data. Finally, there is not just Sentinel-1a/b but there is PALSAR, TerraSAR-X, Tandem-L, COSMO-Skymed an-dRADARSAT / RCM SAR - all currently in orbit and functioning ... hence there is ample of such high-resolution data available - sitting in the archives to be used. Statements such as more sensors "onboard more satellites" should be very well thought about and backed-up by an appropriate analysis - an precondition which is not given by this paper.
» The reviewer is again using the term "smart". See our previous point on how such adjectives should be avoided when one is a reviewer. We anyway modified these sentences.

Editorial remarks

I find usage of "early warning system" not overly appropriate. There is not danger or damage or anything involved. It is too figurative in my eyes and I suggest to write something along the lines that you aim to better understand processes that precondition opening of the Weddell Sea polynya and that you are after to better predict its opening by means of a set of proxies derived from satellite remote sensing.

» We agree with the reviewer and have removed this term from the manuscript.

Line 7 and later in the manuscript, e.g. in section 2.3: "brightness temperature" is a term used in satellite microwave radiometry as well. Therefore I recommend to write "infrared brightness temperature" all the time in your manuscript you are referring to the AVHRR TIR data.

» We now refer to "infrared brightness temperature" throughout the manuscript, including in the figure captions.

» The other comments have already been addressed in response to the previous comments and/or that of the other two reviewers.

---

## Author Comment (AC3) · 5 Feb 2021

We thank the reviewer for their comments. The role of the reviewer has been duly acknowledged with the addition of this sentence in the acknowledgement section:

"We also thank the two anonymous reviewers 1 and 3 and Stephan Kern (reviewer 2) for their comments that greatly improved the clarity and quality of this manuscript"

R: The manuscript has serious weaknesses and is largely not reproducible. First and foremost, the aim of the study is unclear to me. Is it about improving remote sensing methodology or a forecasting system for the opening of the Weddell Polynya? Or both

at the same time? The not very scientific approach and the sloppy writing is bothering (e.g. "debate is closed", "infrared out of fashion").

» We are sorry to hear that the reviewer could not reproduce our results. We have now made our codes freely available on Github, and explained in the Code availability section. In response to reviewer 1's suggestion, we also made a flowchart to clarify our methods. Finally, sections 2 and 3 have been dramatically rewritten (and in the case of section 3, further split into sections 3 and 4) to increase their clarity. Following the reviewer's comment, our aim and working hypothesis are now clearly in the introduction and in the abstract. We do not understand what the reviewer means by "not very scientific" without more specific examples, but we have added a lot of information in response to reviewers 1 and 2. Colloquial sentences have been removed.

R: There is a lack of hypotheses, statistical tests for the significance and descriptions of the uncertainty. References are used in the wrong context and much of the existing literature is ignored. The description of the data is careless and incorrect in some places. If the goal were to analyze the AVHRR data using a new methodology, the question of cloud cover would first have to be analyzed more thoroughly. It remains e.g. unclear whether passive microwaves and AVHRR data give consistent results. This would be the first step towards a suitable long-term study.

» The working hypothesis is now clearly stated in the abstract and in the introduction. References cannot be corrected without more specific examples. We had preferred restricting the literature to that which is most relevant for increased readability, but have added the model-based studies suggested by the reviewer, to provide the wider context. Likewise, data description cannot be corrected without more specific information as to which is wrong. We suspect that our response to the comments from reviewer 1 and 2 and corresponding text modifications address this point. Finally, in response to the reviewer's suggestion as well as that of reviewer 2, a detailed analysis of the cloud cover has been added as appendix section A. In that appendix, we validate the (published) methods that we used for cloud masking against a reference cloud mask

product (MYD35_L2). We also added cloud cover information in Table B1 and added supplementary figure B1, to show that the cloud cover during and in the days leading to a polynya were within the usual "cloudiness" range of the region under non-polynya conditions.

R: An investigation into the causes is difficult to do with observational data alone. The Weddell Polynya is a phenomenon in the coupled system of ocean-ice-atmosphere and is based on feedback effects and tides. Forcings such as fresh water fluxes through precipitation or melting of meteoric ice and the heat reservoir in the deep ocean play a role. Without a coupled model system, causal research is inadequate or must remain empirical. The empirical aspects of the study are however not solid because of the lack of hypotheses, statistical testing and significance.

» We agree with the reviewer that the causes are difficult to investigate from observations alone as the observations are limited in this part of the world, both in terms of resolution and coverage. Using models is what the lead author and her colleagues normally do, for this exact reason. However here it would be beyond the scope of this paper. For once, we wanted to see how much could be achieved from observations alone. We have already addressed the specific reviewer's comment in response to their previous comment.

R: However, the topic is very suitable for the journal and I suggest that the author should resubmit the work after a major revision. Because my concerns are about the main aim of the study I would encourage the author to withdraw the study and to resubmit it with better defined scopes, e.g. in two parts. First a validation of the AVHRR approach to detect the polynya, and a second part about the forecast method. The first part shall include a thorough error analysis about the influence of clouds. The second part should use advanced statistical methods.

» We thank the reviewer for their suggestion. In this revised version, we now mostly focus on the AVHRR approach. As suggested, we include a specific appendix validating

the cloud masking, along with extra figures and tables providing information about the cloud cover. We do not understand what the reviewer means with advanced statistical methods, considering that the reviewer correctly highlighted the limits of observational data. Advanced statistical methods that we commonly use on model output would unfortunately be irrelevant and insignificant here.

---

## Author Response (AR2)

We thank the reviewer for having agreed to review our manuscript a second time. We re-wrote section 4 in a way we consider much clearer, with one subsection per (explicitly stated) hypothesis. We also added more in-situ observations so that section 4 contains fewer suggestions and instead a majority of findings. We nevertheless modified the title, as recommended.

In the following, the reviewer's comment is highlighted with **bold fonts** and our response is in plain text.

**The manuscript has improved but I am still not convinced that the scientific results obtained are sound. Maybe, I have not fully understood the approach. I am confused by the structure and the presented example cases and graphs. Shouldn't the IR test be applied on the full time series, not only on the pre-defined polynya events?**

They are applied to the full time series. See section 3.2.

**The manuscript is not easy to read. The structure with some scientific motivation only in the discussion is not optimal. The hypothesis is not well introduced.**

We mention more clearly our hypotheses throughout the manuscript. In particular, we have entirely re-written section 4 to increase clarity.

**What can be inferred from the detectability about the causes?**

This is the topic of section 4. We have re-written this section to make the connection clearer.

**While the authors agree that it is difficult to investigate the causes with observational data alone they have this aim still prominently in the title of the manuscript. The abstract does not keep what the title promises. The results about the reasons are only very vague**
**"could indicate upwelling of warm water, .. which could indicate a lead..", "may be caused". The strong title and the weak results are thus in contradiction, it remains a speculation about the causes.**

We have modified the title.
We added more in-situ observations to section 4 for our conclusions to be more robust. Two co-authors have been added to the manuscript who helped with these new analyses.
Bearing in mind the reviewer's comment on the previous version of the manuscript, we also conducted extra analyses that allowed us to compute correlations and actual quantifications, instead of the visual description of the original manuscript.

**The writing style has unfortunately not much improved. For example: why use adjectives like "painfully"? In the next sentence why use the word "warning"? Taken together this suggests that the polynya is something that could cause harm or danger.**

We see from the reviewer's comments that our sentences were misunderstood. We removed the words that the reviewer criticised.

**Important comments from reviewer 2 have not been considered.**

To the best of our knowledge, we have addressed all of reviewer 2's comments. Please indicate which comments you are referring to.

**It is unclear why navigation is used in the first sentence to motivate the study.**

We removed this word. There genuinely are plans for commercial exploitation of ice-infested waters, in particular of polynyas, but mentioning them is beyond the scope of this paper.

**I disagree with the statement that the Polynya re-appeared in 2016 for the first time in forty years. There was at least a report about the reappearance of the winter Weddell Polynya in 1994, Drinkwater (1998). And the halo of the reduced ice concentration over Maud Rise was very often visible before and well documented in the literature as described in the manuscript. Thus, if it is written "for the first time" the criteria has to be explained.**

What the reviewer describes is our entire section 3.1. We rephrased the introduction.

**Page 2**
**"SAR is a comparatively recent technology for sea ice observation, providing climate data only since the early 2000s" - Seasat provided SAR data already in 1978. What "climate" data sets have been derived from SAR? I don't know any.**

This sentence has been removed.

**Page 3**
**25 km resolution is the grid resolution of the product, not the spatial resolution of the sensor.**

We modified this sentence to avoid such confusion.

**Comments from the Editor**

We thank the editor for their comments. We added their contribution to the acknowledgement section.

**Specific points;**
**All line numbers refer to the track change manuscript.**

**Abstract line 4: either "the cause of the opening is not yet determined" or "the cause of the opening is much debated".**

We rephrased.

**I would caution you not to use T3b, T4 and T5 in the abstract as this is jargon not many readers may understand.**

We removed references to T3b, T4 and T5 from the abstract

**Page 4 Line 124: "so not for us" is colloquial.**

We rephrased

**Page 6 line 134: "is only directly available"**

We rephrased.

**Page 6 line 165: "so-called" is colloquial.**

Here, sincerely, thank you. I had never been told that it was colloquial / negative. We removed it from the manuscript (and from the proposal we were about to submit when your comment reached us)

**Page 16 line 287: "could also be warm air"**

This sentence has been removed from the manuscript.

**line 298: "snow layer" duplicated**

This sentence has been removed from the manuscript.

**Page 19: line 398: "so it is more likely that the"**

We do not see which sentence this comment refers to – words, page and line numbers do not match.

**line 399, this sentence is awkward**

Likewise, we do not see which sentence this comment refers to – line 399 on page 21 is entirely crossed out and the surrounding sentence does not sound awkward. Could you please suggest an alternative?

---

## Author Response (AR3)

**Response to the Editor's comments**

We thank the Editor for their comments. All their comments have been addressed, resulting in some rewriting to increase clarity and the addition of one table to section 4.1 (Table 1).

The Editor's comments are in **bold font**; our reply, in plain font.

**You have developed a method to forecast polynya opening over Maud Rise based, and have demonstrated that this forecast does not have false positives in the satellite record. It is also useful to know if there are false negatives, and what the occurance of this is. Looking at figure 5 I think there maybe some small overlap where the standard deviation in brightness temperature of the three channels is similar to climatology for some polynya events. Can you quantify this?**

Figure 5, especially the bottom scatter plots, shows that there is no overlap between the value in the climatology and that in polynya years. We further verified this result when we first obtained it months ago, and consequently explicitly indicated in the manuscript already then "We find no false positive" (line 202 of the new, non-track-changes manuscript). This finding is indicated again in the conclusions: "…successfully detected the events without finding false positives" (line 351).

We removed the confusing sentence about "minimising the amount of false positives" from the introduction.

**I am concerned about the additional analysis presented in section 4, where you are exploring if AVHRR can be used to identify physical features of the upper ocean or ice that may be associated with polynya opening. In particular, I do not see how the surface brightness temperature can be related directly to ocean properties under the ice in winter. See my simple reasoning for this in the specific comments below. The only finding you have is that there is correspondence with wind curl suggesting T4 decreases under opening favorable winds and T45 is correlated to opening. I think the only thing you can really say out of this is that the brightness temperature changes associated with polynya opening are probably just showing the lead opening that is a precursor to polynya forming. It is very important that you are clear about the limitations of your findings here. Really there is no indication that AVHRR can tell you anything more than that there is a change in ice concentration. My take home message from this is that AVHRR and other satellite remote sensing have to be used together with oceanographic data to identify mechanisms causing the polynya to open. Section 4 needs a tighter focus to ensure readers are not misled that you can say anything about the interior ocean under ice with AVHRR. Your finding appears to be that you might be able to identify opening, which might be a precursor to the polynya open, but can not say anything about mechanisms creating sensible heat polynyas. This is an important distinction if it is the case. What I get out of this section is somewhat of an understanding of why the forecast method might work, and that it might be limited to polynya's that require wind driven opening as a precursor to their formation. Thinking this way refocusses section 4 away from over-reach to try to identify mechanisms of polynya formation to understanding why the forecast might work for this particular polynya. Which does make my question in the paragraph above important, as we could ask does every detected reduction in variability of the various brightness temperature measures (which appears to be related to wind driven opening) result in a polynya?**

Our working hypothesis is that the behaviour of the ocean in the subsurface, and in particular its vertical motion of comparatively warm water, affects the water column up to the surface. As the

surface is affected, it impacts the sea ice: it either opens leads, or it thins the ice. Note here that we do not necessarily assume a change in sea ice concentration; a change in sea ice thickness would matter as much.

The first step is to upwell deeper waters, here the comparatively warm Circumpolar Deep Water, to the surface. Such upwelling to the surface is a common process, all around the world; in nutrient-rich regions, it even is clearly detectable in visible satellite imagery. What it means at the latitudes of our study area is that the comparatively warm deep water, with a core at 1 deg C or more (Fig 7), when upwelled, brings waters above freezing temperature in contact with the sea ice. This is not shown in the mooring series of Fig 7 because there are no sensors at the surface due to the risk of iceberg collision, but it has been reported previously (e.g. Anderson, 1961; McPhee, 1992; Jacobs et al., 2012; Timmermans, 2015) . Besides, the Circumpolar Deep Water is comparatively salty (supp. Fig B3), so its upwelling increases the ocean surface salinity. There are then at least three different ways through which CDW upwelling affects the sea ice:

1) A thin enough ice may open leads in response to the upwelling, resulting in a heat loss from the ocean to the atmosphere. In our understanding, the Editor argues that upwelling and leads are both the result of wind rather than cause and effect; past literature (already cited throughout our manuscript, see e.g. review by Campbell et al., 2019) has shown on several occasions that at least in the case of the Weddell/Maud Rise polynya, wind alone is not enough to open the ice and that upwelling is required, i.e. upwelling is a cause.

2) Increased ocean surface temperature melts the sea ice bottom and/or results in increased conduction of heat through the ice (e.g. McPhee and Untersteiner, 1982); we here ignore the enhancing effect of momentum flux (McPhee, 1992), as we have no information about the ice bottom topography.

3) Increased ocean surface salinity results in increased convection through the ice brine channels (e.g. Lytle and Ackley, 1996). Also, this increase in salinity lowers the freezing temperature, further enhancing the second effect.

All three processes result in heat and moisture loss from the ocean, which is what we argue we see on the AVHRR data as an increase in brightness temperature.

We acknowledge that there is an extra, technical challenge in the fact that brightness temperature retrieval from antenna count will be impacted by more processes than what we just described, notably changes in cloud cover, in air temperature, wind speed, humidity, and the ice emissivity and scattering (e.g. Scambos et al., 2006; Bushuev et al., 2007).

Therefore, to answer this major comment along with the specific comment of lines 251-254, we added a shortened version of the scientific motivation of our working hypothesis to the beginning of section 4.1. We also changed the title of subsection 4.1, as suggested by the Editor.

We also added a new table (Table 1) and the discussion of that table to section 4.1. That table shows the correlation between T4 and the ocean temperature from the mooring, but also the air temperature, the relative humidity, and the wind speed from ERA5, in the 15 days leading to each polynya event for the events that occurred after 1996 (deployment of the moorings). Cloud cover changes are already accounted for as we masked the cloud pixels. As we lack adapted data for sea ice emissivity and scattering, and these two properties are not independent from air temperature and wind speed, we decided to not study them further. What this new table shows is that for most events, the correlation is strongest between T4 and the ocean data, but that there are indeed instances where

our results are inconclusive. In the abstract, the end of section 4.1, and the conclusions, we now clearly indicate that other processes impact the brightness temperature retrieval, and that therefore T4 should not be used as a proxy without caution.

**I highly recommend you read the literature on detecting leads in AVHRR. Here is a starting point: Lindsay, R. W., & Rothrock, D. A. (1995). Arctic sea ice leads from advanced very high resolution radiometer images. Journal of Geophysical Research: Oceans, 100(C3), 4533-4544. You should also consider what the brightness temperature is telling you about the skin temperature in the sensor footprint. There is also a ice surface product based on MODIS, that uses similar information.**
**Lindsay, R. W., & Rothrock, D. A. (1994). Arctic sea ice surface temperature from AVHRR. Journal of Climate, 7(1), 174-183.**

**Hall, D. K., Key, J. R., Casey, K. A., Riggs, G. A., & Cavalieri, D. J. (2004). Sea ice surface temperature product from MODIS. IEEE transactions on geoscience and remote sensing, 42(5), 1076-1087. It will greatly help the manuscript if you are clear about what the values you use in your forecast are actually measuring.**

We thank the Editor for the recommendations. We are aware of these papers and cited them in a previous version of this manuscript. The main issue we have with them is that, as often, they assume that the ocean is at freezing temperature. As explained above, it is not necessarily the case.

In general, we are wary of empirical retrievals using coefficients computed using in-situ data from a different part of the world, and that is why we chose to work with brightness temperature rather than skin temperature.

**I have some specific comments. Line numbers are based on the track change version of the manuscript.**

**Title: I would suggest you do not need 'upcoming' in the title, you are already pointing to early detection.**

"Upcoming" has been removed from the title.

**line 7: "we find in fact 30 polynyas". I would remove "in fact".**

Changed as suggested.

**Line 110-115 (end of section 2.1): The float surfacing indicates open water. A lead is likely to have water at the freezing point, and will only have water close to the surface in summer at higher than freezing. Also, I am not understanding the last sentence in this paragraph. The float tries to surface every 10 days. If it surfaces twice this indicates open water on both those dates, but not that there was open water 10 days prior to when it surfaced.**

By design, the float does not surface if the water is at freezing temperature. More specifically, it aborts the ascent if the median temperature between 50 and 20 m depth is lower than -1.78 degC, and flags the profile as "ice detected". To allow surfacing, the algorithm requires two consecutive "ice not detected" profiles, i.e. two consecutive profiles with above freezing temperatures. The first above freezing / "ice not detected" profile will not result in a full ascent, so we are not certain that there was open water then. We have rephrased these sentences to clarify what happens on the first ascent.

**The sea ice concentration time series is not referenced where you first introduce it. Also it is my understanding that the NSIDC and Comiso bootstrap algorithms are different. In the conclusion you appear to refer to the NSIDC data set as the bootstrap data. Please clarify and make sure the appropriate data citation is included in the paper.**

The sea ice concentration time series is introduced as early as the Introduction section, as reviewer 2 had requested two revisions ago. We rephrased throughout the manuscript to clarify that it is the Comiso bootstrap algorithm, and that the dataset was downloaded from the NSIDC website. The appropriate data reference is given in the Introduction section, and the product doi is indicated in the first paragraph of section 2 and in the data availability section.

**line 225, you do not need the second question mark in this paragraph.**

Changed as suggested.

**line 251-254: "hypothesised that oscillations in the infrared brightness temperature, especially in T4, in the days before the polynya opens might reflect oceanic convective movements. Their argument is that as the warm water is being upwelled, more heat is going through the ice." The base of the ice is always at freezing point. So it is salinity changes that affects the temperature of the base. This will vary very little compared to the top surface. The heat flux through the ice is indeed from warm bottom to cold top surface prior to the onset of surface melt, but the rate will be influenced more by the atmospheric temperature than the ocean temperature. The increase in brightness temperature can be due to enhanced open water, so it would be very hard to dissociate the effect of opening from upwelling, and based on my reasoning above I actually think any upwelling signal would be much smaller than that due to changes in the ice thickness within the satellite footprint. I do think the mooring analysis is interesting, and the correspondence of opening and upwelling (or eddy transfer of warm water) is interesting. In the context of the remote sensing I am not convinced that the variability in T4 is associated with the water property changes. Your discussion is suggesting the same thing, that the signal is likely leads opening. I feel it could be more clearly pointed out that correlation is not causation, and the upwelling/eddy presence is happening at the same time as ice opening. I actually do not understand how you might be able to detect ocean warming under the ice in the skin temperature. This is possible in the open ocean, but if there is ice present the temperature changes at the surface due to heat flux change from the ocean will be much more subtle and maybe lower than the resolution of the sensor. I suggest the title of this section (4.1) is misleading, because you do not show that variation in T4 is directly related to upwelling, or that there is even a statistically significant relationship. In fact you even say "The T4 data are too patchy to robustly determine the increase in brightness temperature that corresponds to upwelling". What your findings do indicate is summed up in lines 312-314 that there is a slight negative correlation between wind favorable to upwelling and surface temperature change (temperature reducing). And you point out that the mooring data shows warm/salty water closer to the surface prior to polynya events, but these might be eddies rather than wind driven upwelling. Essentially the oceanographic data is suggesting that there should not be a direct correspondence between upper ocean warming and ice opening, which is what you show. My feeling is that it is pretty obvious you can not use skin temperature to determine what is happening at depth in the ice covered ocean in winter. It is possible that you need both warm water and wind driven opening to create a polynya and these are not necessarily happening at the same time, it might just be the coincidence of a warm core eddy passing as the ice opens.**

This comment has already been answered at lengths in response to the Editor's second major comment. We would like to insist again that although the base of the ice is assumed to be at freezing temperature for most retrievals, it is not necessarily true. Besides, salinity changes are associated with temperature changes, as the upwelled CDW is not only warm but also salty.

As indicated in response to major comment 2, to answer this specific comment along with major comment 2, we added a shortened version of the scientific motivation of our working hypothesis to the beginning of section 4.1. We also changed the title of subsection 4.1, as suggested by the Editor. We also added a new table (Table 1) and the discussion of that table to section 4.1. That table shows the correlation between T4 and the ocean temperature from the mooring, but also the air temperature, the relative humidity and the wind speed from ERA5, in the 15 days leading to each polynya event for the events that occurred after 1996 (deployment of the moorings). Cloud cover changes are already accounted for as we masked the cloud pixels. As we lack adapted data for sea ice emissivity and scattering, and these two properties are not independent from air temperature and wind speed, we decided to not study them further. What this new table shows is that for most events, the correlation is strongest between T4 and the ocean data, but that there are indeed instances where our results are inconclusive. In the abstract, the end of section 4.1, and the conclusions, we now clearly indicate that other processes impact the brightness temperature retrieval, and that therefore T4 should not be used as a proxy without caution.

**Line 278: "On the three moorings, all hydrographic sensors are in the CDW (symbols on Fig. 7 and supplementary Table B2), but the shallowest ones are above the core and can therefore be used to observe potential upwelling." Confusing, you first say there are no sensors outside the CDW then you say there are.**

This sentence distinguishes between the overall CDW layer and its core, i.e. the depth level of the temperature and salinity maxima. So what we say (and show) is that there are no sensors outside the overall CDW layer. We then say, based on Figs 7 and B3, that there are sensors outside the core of the CDW. As temperature and salinity decrease away from the core, increases in temperature and salinity detected by these sensors above the core indicate an upwelling.  We re-phrased to increase clarity.

**line 299, I think you need a comma after "In the polynya years we detected"**

Modified as suggested.

**line 313: "lead opening", I would caution you just to say opening of the ice, for the reasons stated above.**

Rephrased as suggested.

**line 319" We now investigate whether these leads can be detected" might be "We now investigate whether this ice opening can be detected" ... and in other places**

Rephrased as suggested.

**Line 415: "The next step would be to check whether these criteria are still valid for other Antarctic open ocean polynyas,but also coastal polynyas(e.g. the Amundsen Sea Polynya, Randall-Goodwin et al., 2015) or even Arctic polynyas (e.g. North Water and North Green-land polynyas, Preußer et al., 2015; Ludwig et al., 2019)." "but also" should be "including". I would also remove "even" from "even Arctic polynyas".**

Coastal polynyas are not a subset of open ocean polynyas, so the suggested change to "including" would be incorrect. We rephrased this sentence.

**Figures: You will want to take a careful review of these. Figure 3 has part of a bounding box visible around the map that overlaps a color bar title. Figure 2 has labels that are more designed for a slide show than a publication, consider if a legend would be appropriate rather than the boxed labels.**

We struggled to reproduce the issue that the Editor had with Figure 3, and eventually found that it was pdf-reader specific. Our solution for now has been to replace all pdf figures with a png version, which results in lower resolution. Shall this manuscript be accepted, we will work with the copy editors to ensure that the high quality pdf version displays correctly for all pdf-readers.

We modified Figure 2 as suggested.

**Data citations are required for the AVHRR and NSIDC data used in the paper. Please make sure these are provided in the acknowledgements.**

Data citations are provided throughout the manuscript, including the dedicated Data availability section. We have now explicitly added the dataset doi of these two products to that section.